# Trained Models Tell Us How to Make Them Robust to Spurious Correlation without Group Annotation

## Abstract

Classifiers trained with Empirical Risk Minimization (ERM) tend to rely on attributes that have high spurious correlation with the target. This can degrade the performance on underrepresented (or *minority*) groups that lack these attributes, posing significant challenges for both out-of-distribution generalization and fairness objectives. Many studies aim to improve robustness to spurious correlation, yet nearly all require group annotation for training and/or model selection. This constrains their applicability in situations where the nature of the spurious correlation is not known, or when group labels for certain spurious attributes are either insufficient or completely absent. To meet the demand for effectively enhancing the model robustness under minimal assumptions about group annotation, we propose Environment-based Validation and Loss-based Sampling (EVaLS). It uses the losses from a trained model to construct a balanced dataset of high-loss and low-loss samples in which the training data group imbalance is mitigated. This results in a significant robustness to group shifts when equipped with a simple mechanism of last layer retraining. Furthermore, by utilizing environment inference methods for creating diverse environments with correlation shifts, EVaLS can potentially eliminate the need for group annotation in the validation data. In such a context, the worst environment accuracy acts as a reliable surrogate throughout the retraining process for tuning hyperparameters and finding a model that performs well across diverse group shifts. EVaLS effectively achieves group robustness, showing that group annotation is not necessary even for validation. It is a fast, straightforward, and effective approach that reaches near-optimal worst group accuracy without needing group annotations, marking a new chapter in the robustness of trained models against spurious correlation.

## 1 Introduction

Training deep learning models using Empirical Risk Minimization (ERM) on a dataset, poses the risk of relying on *spurious correlation*. These are correlations between certain patterns in the training dataset and the target (e.g., the class label in a classification task) despite lacking any causal relationship. Learning such correlations as shortcuts can negatively impact the models' accuracy on *minority groups* that do not contain the spurious patterns associated with the target [1, 2]. This problem leads to concerns regarding fairness [3], and can also cause a marked reduction in the performance. This occurs particularly when minority groups, which are underrepresented during training, become overrepresented at the time of testing, as a result of shifts within the subpopulations [4]. Hence, ensuring robustness to group shifts and developing methods that improve *worst group accuracy* (WGA) is crucial for achieving both fairness and robustness in the realm of deep learning.

Submitted to 38th Conference on Neural Information Processing Systems (NeurIPS 2024). Do not distribute.

Many studies have proposed solutions to address this challenge. A promising line of research focuses on increasing the contribution of minority groups in the model's training [1, 5–7]. A strong assumption that is considered by some previous works is having access to group annotations for training or fully/partially fine-tuning a pretrained model [8, 7, 1]. The study by Kirichenko et al. [1] proposes that retraining the last layer of a model on a dataset which is balanced in terms of group annotation can effectively enhance the model's robustness against shifts in spurious correlation. While these works have shown tremendous robustness performance, their assumption for the availability of group annotation restricts their usage.

In many real-world applications, the process of labeling samples according to their respective groups can be prohibitively expensive, and sometimes impractical, especially when all minority groups may not be identifiable beforehand. A widely adopted strategy in these situations involves the indirect inference of various groups, followed by the training of models using a loss function that is balanced across groups [5, 9, 10, 4]. The loss value of the model or its similar metrics is a popular signal for recognizing minority groups [5, 9–11]. While most of these techniques necessitate full training of a model, Qiu et al. [9] attempt to adapt the DFR method [1] with the aim of preserving computational efficiency while simultaneously improving robustness to group shift. However, this method still requires group annotations of the validation set for model selection and hyperparameter tuning. Consequently, this constitutes a restrictive assumption when adequate annotations for certain groups are not supplied. It also applies to situations where some shortcut attributes are completely unidentified.

In this study, we present a novel strategy that effectively mitigates reliance on spurious correlation, completely eliminating the need for group annotations during both training and retraining. More interestingly, we provide empirical evidence indicating that group annotations are not necessary, even for model selection. We show that assembling a diverse collection of environments for model selection, which reflect group shifts can serve as an effective alternative approach. Our proposed method, Environment-based Validation and Loss-based Sampling (EVaLS), is a technique that strengthens the robustness of trained models against spurious correlation, all without relying on group annotations. EVaLS is pioneering in its ability to eliminate the need for group annotations at *every phase*, including the model selection step. EVaLS posits that in the absence of group annotations, a set of *environments* showcasing group shifts is sufficient. Worst Environment Accuracy (WEA) could then be utilized for model selection. Our findings demonstrate that utilizing environment inference methods [12], or even dividing the validation data based on the predictions of a random linear layer atop a trained model's feature space can markedly enhance group robustness. Figure 1 demonstrates the overall procedure of the main parts of EVaLS.

Our empirical observations support prior research which suggests that high-loss data points in a trained model may signal the presence of minority groups [5, 9, 10]. Our method, EVaLS, evenly selects from both high-loss and low-loss data to form a balanced dataset that is used for last-layer retraining. We offer theoretical explanations for the effectiveness of this approach in addressing group imbalances, and experimentally show the superiority of our efficient solution to the previous strategies. Comprehensive experiments conducted on spurious correlation benchmarks such as CelebA [13], Waterbirds [7], and UrbanCars [14], demonstrate that EVaLS achieves optimal accuracy. Moreover, when group annotations are accessible solely for model selection, our approach, EVaLS-GL, exhibits enhanced performance against various distribution shifts, including attribute imbalance, as seen in MultiNLI [15], and class imbalance, exemplified by CivilComments [16]. We further present a new dataset, *Dominoes Colored-MNIST-FashionMNIST*, which depicts a situation featuring multiple independent shortcuts, that group annotations are only available for part of them (see Section 2.2). In this setting, we show that strategies with lower levels of group supervision are paradoxically more effective in mitigating the reliance on both known and unknown shortcuts.

The main contributions of this paper are summarized as follows:

- We present EVaLS, a simple yet effective approach that enhances model robustness against spurious correlation without relying on ground-truth group annotations.

- We offer both theoretical and practical insights on how balanced sampling from high-loss and low-loss samples can result in a dataset in which the group imbalance is notably mitigated.

- Using simple environment inference techniques, EVaLS leverages worst environment accuracy as a reliable indicator for model selection.

- EVaLS attains near-optimal worst group accuracies or even exceeds them in spurious correlation benchmarks, all with zero group annotations.

- When group annotations are available for model selection, EVaLS delivers state-of-the-art performance across a variety of subpopulation shift benchmarks.

- We introduce a new dataset consisting of two spurious features in which partial supervision may negatively impact the performance of the underrepresented groups.

## 2 Preliminaries

### 2.1 Problem Setting

We assume a general setting of a supervised learning problem with distinct data partitions $\mathcal{D}^{\text{tr}}$ for training, $\mathcal{D}^{\text{val}}$ for validation, and $\mathcal{D}^{\text{test}}$ for final evaluation. Each dataset comprises a set of paired samples $(x, y)$, where $x \in \mathcal{X}$ represents the data and $y \in \mathcal{Y}$ denotes the corresponding labels. Conventionally, $\mathcal{D}^{\text{tr}}$, $\mathcal{D}^{\text{val}}$, and $\mathcal{D}^{\text{test}}$ are assumed to be uniformly sampled from the same distribution. However, this idealized assumption does not hold in many real-world problems where distribution shift is inevitable. In this context, we consider the sub-population shift problem [4]. In a general form of this setting, it is assumed that data samples consist of different groups $\mathcal{G}_i$, where each group comprises samples that share a property. More specifically, the overall data distribution $p(x, y) = \sum_i \alpha_i p_i(x, y)$ is a composition of individual group distributions $p_i(x, y)$ weighted by their respective proportions $\alpha_i$, where $\sum_i \alpha_i = 1$. In this work, we assume that $\mathcal{D}^{\text{tr}}$, $\mathcal{D}^{\text{val}}$, and $\mathcal{D}^{\text{test}}$ are composed of identical groups but with a different set of mixing coefficients $\{\alpha_i\}$. It is noteworthy that the validation set may have approximately identical coefficients to those of the training or testing sets, or it may have entirely different coefficients.

Several kinds of subpopulation shifts are defined in the literature, including class imbalance, attribute imbalance, and spurious correlation [4]. Class imbalance refers to the cases where there is a difference between the proportion of samples from each class, while attribute imbalance occurs when instances with a certain attribute are underrepresented in the training data, even though this attribute may not necessarily be a reliable predictor of the label. On the other hand, spurious correlation occurs when various groups are differentiated by spurious attributes that are partially predictive and correlated with class labels but are causally irrelevant. More precisely, we can consider a set of spurious attributes $\mathcal{S}$ that partition the data into $|\mathcal{S}| \times |\mathcal{Y}|$ groups. When the concurrence of a spurious attribute with a label is significantly higher than its correlation with other labels, that spurious attribute could become predictive of the label, resulting in deep models relying on the spurious attributes as shortcuts instead of the core ones. This is followed by a decrease in the model's performance on groups that do not have this attribute.

Given a class, the group containing samples with correlated spurious attributes is referred to as *majority* group of that class, while the other groups are called the *minority* groups. As an example, in the Waterbirds dataset [7], for which the task is to classify images of birds into landbird and waterbird, there are spurious attributes $\{water\ background, land\ background\}$. Each background is spuriously correlated with its associated label, decompose the data into two majority groups *waterbird on water background*, and *landbird on land background*, and two minority groups *waterbird on land background* and *landbird on water background*. Our goal is to make the classifier robust to spurious attributes by increasing performance for all groups.

### 2.2 Robustness of a Trained Model to an Unknown Shortcut

In scenarios where group annotations are absent, traditional methods that depend on these annotations for training or model selection become infeasible. Moreover, as previously discussed by [14], when data contains multiple spurious attributes and annotations are only available for some of them, such methods would make the model robust only to the known spurious attributes. To explore such complex scenarios, we introduce the *Dominoes Colored-MNIST-FashionMNIST (Dominoes CMF)* dataset (Figure 3(d)). Drawing inspiration from Pagliardini et al. [17] and Arjovsky et al. [18], Dominoes CMF merges an image from CIFAR10 [19] at the top with a colored (red or green) MNIST [20] or FashionMNIST [21] image at the bottom. The primary label is derived from the CIFAR10 image, while the bottom part introduces two independent spurious attributes: color and shape. Although

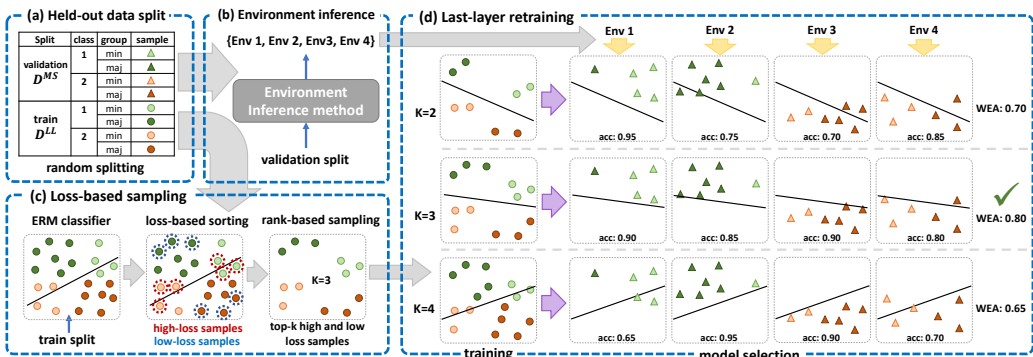

Figure 1: Overview of the proposed method. Given an ERM-trained model (similar to DFR [1]), the following steps are performed: (a) we randomly split the held-out dataset into train and validation splits. (b) An environment inference method is utilized to infer diverse environments from the validation split. (c) We evaluate train split samples on the initial ERM classifier and sort high-loss and low-loss samples of each class for loss-based sampling. (d) Finally, we perform last-layer retraining on the loss-based selected samples. Each retraining setting (e.g. different $k$ for loss-based sampling) is validated based on the worst accuracy of the inferred environments. Note that majority and minority groups are shown with dark and light colors for better visualization, but are not known in our setting.

annotations for shape are provided for training and model selection, color remains an unknown variable until testing. For more details on the dataset refer to the Appendix.

The illustrations in Figure 3(a-c) depict the outlined scenario. A classifier trained using ERM is dependent on both spurious features (Figure 3(b)). Yet, achieving robustness against one spurious correlation (Figure 3(c)), does not ensure robustness against both (Figure 3(a)). In Section 4 we show that our method, which does not rely on the group annotations of the identified group, achieves enhanced robustness against both spurious correlations, outperforming strategies that depend on the known group's information.

## 3 Environment-based Validation and Loss-based Sampling

Our method, EVaLS, is designed to improve the robustness of deep learning models to group shifts without the need for group annotation. In line with the DFR [1] approach, we utilize a classifier defined as $f = h_\phi \circ g_\theta$, where $g_\theta$ represents a deep neural network serving as a feature extractor, and $h_\phi$ denotes a linear classifier. The classifier is initially trained with the ERM objective on the training dataset $\mathcal{D}^{tr}$. Subsequently, we freeze the feature extractor $g_\theta$ and focus solely on retraining the last linear layer $h_\phi$ using the validation dataset $\mathcal{D}^{\text{val}}$ as a held-out dataset.

We randomly divide the validation set $\mathcal{D}^{\text{val}}$ into two subsets, $\mathcal{D}^{LL}$ and $\mathcal{D}^{MS}$ which are used for last layer training and model selection, respectively. In Section 3.1 we explain how to sample a subset of $\mathcal{D}^{LL}$ that statistically handles the group shifts inherent in the dataset. In Section 3.2 we describe how $\mathcal{D}^{MS}$ is divided into different environments that are later used for model selection. The optimal number of selected samples from $\mathcal{D}^{LL}$ and other hyperparameters is determined based on the worst environment accuracies among environments that are obtained from $\mathcal{D}^{MS}$. By combining our novel sampling and validation strategy, we aim to provide a robust linear classifier $h_{\phi^*}$ that significantly improves the accuracy of underrepresented groups without requiring group annotations of training or validation sets. Figure 1 illustrates the comprehensive workflow of the EVaLS methodology. Finally in Section 3.3, we provide theoretical support for the loss-based sampling procedure and its effectiveness.

### 3.1 Loss-Based Instance Sampling

Following previous works [5, 10, 9], we use the loss value as an indicator for identifying minority groups. We first evaluate classifier $f$ on samples within $\mathcal{D}^{LL}$ and choose $k$ samples with the highest and lowest loss values for a given $k$. By combining these $2k$ samples from each class, we construct a

balanced set $\mathcal{D}^{balanced}$, consisting of high-loss and low-loss samples (see Figure 1(c)). $\mathcal{D}^{balanced}$ is then used for the training of the last layer of the model.

As depicted in Figure 2, the proportion of minority samples among various percentiles of samples with the highest loss values increases as we select a smaller subset of samples with the highest loss. This suggests that high and low-loss samples could serve as effective representatives of minority and majority groups, respectively. In Section 3.3, we offer theoretical insights explaining why this approach could lead to the creation of group-balanced data.

## 3.2 Partitioning Validation Set into Environments

Contrary to common assumptions and practices in the field, precise group labels for the validation set are not essential for training models robust to spurious correlations. Our empirical findings, detailed in Section 4, reveal that partitioning the validation set into environments that exhibit significant subpopulation shifts can be used for model selection. Under these conditions, the worst environment accuracy (WEA) emerges as a viable metric for selecting the most effective model and hyperparameters.

The concept of an *environment*, as frequently discussed in the invariant learning literature, denotes partitions of data that exhibit different distributions. A model that consistently excels across these varied environments, achieving impressive worst environment accuracy (WEA), is likely to perform equally well across different groups in the test set. Several methods for inferring environments with notable distribution shifts have been introduced [12, 22]. Environment Inference for Invariant Learning (EIIL) [12], leverages the predictions from an earlier trained ERM model to divide the data into two distinct environments that significantly deviate from the invariant learning principle proposed by Arjovsky et al. [18], thus creating environments with distribution shifts. Initially, EIIL is employed to split $\mathcal{D}^{MS}$ into two environments. Subsequently, each environment is further divided based on sample labels, resulting in $2 \times |\mathcal{Y}|$ environments. To measure the difference between the distribution of environments, we define *group shift* of a class as the absolute difference in the proportion of a minority group between two environments of that class. A higher group shift suggests a more distinct separation between environments. As detailed in the Appendix, environments inferred by EIIL demonstrate an average group shift of $28.7\%$ over datasets with spurious correlation. Further information about EIIL and the group shift quantities for each dataset can be found in the Appendix.

We demonstrate that even more straightforward techniques, such as applying a random linear layer over the feature embedding space and distinguishing environments based on correctly and incorrectly classified samples of each class, can be effective to an extent in several cases (See Appendix E.2). It underscores that the feature space of a trained model is a valuable resource of information for identifying groups affected by spurious correlations. This supports the logic of previous research that employs clustering [23] or contrastive methods [24] in this space to differentiate between groups.

## 3.3 Theoretical Analysis

In this subsection, we provide theoretical insights into why loss-based sampling in a class can be utilized to create a balanced dataset of each group under sufficient conditions. We will show the close relationship between the existence of a balanced dataset and the difference between the minority vs. majority group means, calculated based on the logits of an ERM-trained classifier. Such logits are known to depend on spurious features. Hence the mentioned group mean difference is expected to be high if spurious features are present in the dataset.

Consider a binary classification problem with a cross-entropy loss function. Let logits be denoted as $L$. Because loss is a monotonic function of logits, the tails of the distribution of loss across samples are equivalent to that of the logits in each class.

We assume that in feature space (output of $g_\theta$) samples from the minority and majority of a class are derived from Gaussian distributions. So, we can consider $\mathcal{N}(\mu_{\min}, \sigma_{\min}^2)$ and $\mathcal{N}(\mu_{\mathrm{maj}}, \sigma_{\mathrm{maj}}^2)$ as the distribution of minority and majority samples in logits space.

**Proposition 3.1** (Feasiblity Of Loss-based Group Balancing). *Suppose that $L$ is derived from the mixture of two distributions $\mathcal{N}(\mu_{min}, \sigma_{min}^2)$ and $\mathcal{N}(\mu_{maj}, \sigma_{maj}^2)$ with proportion of $\varepsilon$ and $1 - \varepsilon$, respectively, where $\varepsilon \leq \frac{1}{2}$. Under sufficient (see App.C) and necessary conditions on $\mu_{min}$, $\mu_{maj}$, $\sigma_{min}$ and $\sigma_{maj}$ including inequality 1, there exists $\alpha$ and $\beta$ such that restricting $L$ to the $\alpha$-left and*

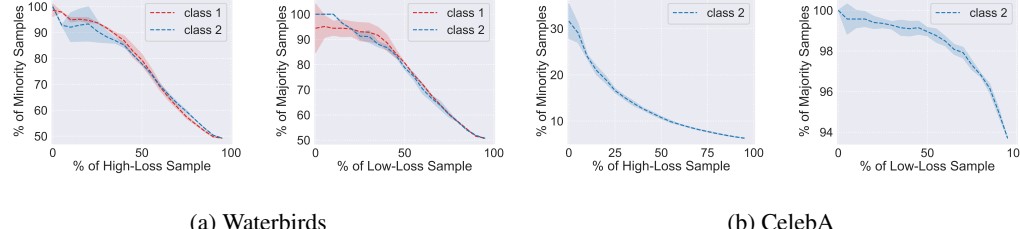

(a) Waterbirds                                    (b) CelebA

Figure 2: The proportion of minority(majority) samples across different classes within various percentages of $\mathcal{D}^{LL}$ samples with highest (lowest) loss for the Waterbirds (a) and CelebA (b) datasets. Minority group samples are more prevalent among high-loss samples, while majority group samples dominate the low-loss areas. The error bars are calculated across three ERM models. [1]

$\beta$-right tails of its distribution results in a group-balanced distribution; in which both components are equally represented.

$$\epsilon \geq \text{sigmoid}\left( -\frac{(\mu_{\text{maj}} - \mu_{\text{min}})^2}{2(\sigma_{\text{maj}}^2 - \sigma_{\text{min}}^2)} - \log\left(\frac{\sigma_{\text{maj}}}{\sigma_{\text{min}}}\right) \right) \tag{1}$$

We provide an outline for proof of Proposition 3.1 here and leave the complete and formal proof and also exact bounds to Appendix C. We also analyze the conditions and effects of spurious correlation in satisfying these conditions. To proceed with the outline we first define a key concept to outline our proof.

**Definition 3.1** (Proportional Density Difference). *For any interval $I = (a, b]$ and a mixture distribution $\varepsilon P_1(x) + (1 - \varepsilon)P_2(x)$, the proportional density difference is defined as the difference of accumulation of two component distributions in the interval $I$ and is denoted by $\Delta_\varepsilon P_{mixture}(I)$.*

$$\Delta_\varepsilon P_{mixture}(I) \overset{\Delta}{=} \varepsilon P_1(x \in I) - (1 - \varepsilon)P_2(x \in I)$$

**Proof outline**    Our proof proceeds with three steps. First, we reformulate the theorem as an equality of left- and right-tail proportional distribution differences. In other words, we show that the more mass the minority distribution has on one tail, the more mass the majority distribution must have on the other tail. Afterward, supposing $\mu_{\text{min}} < \mu_{\text{maj}}$ WOLG, we propose a proper range for $\beta$ values on the right tail. We show that when $\sigma_{\text{maj}} \leq \sigma_{\text{min}}$, values for $\alpha$ trivially exist that can overcome the imbalance between the two distributions. In the last step, for the case in which the variance of the majority is higher than the minority, we discuss a necessary and sufficient condition for the existence of $\alpha$ and $\beta$ based on the left-tail proportional density difference using the properties of its derivative with respect to $\alpha$.

Condition 1 suggests that for a given degree of spurious correlation $\epsilon$ and variations $\sigma_{\text{maj}}, \sigma_{\text{min}}$, an essential prerequisite for the efficacy of loss-based sampling is a sufficiently large disparity between the mean distributions of minority and majority samples, denoted by $\|\mu_{\text{maj}} - \mu_{\text{min}}\|^2$. This indicates that the groups should be distinctly separable in the logits space.

Although the parameters $\alpha$ and $\beta$ are theoretically established under certain conditions, their actual values are undetermined. Therefore, validation data is necessary to ascertain them. For practicality and simplicity in this study, we consider that $\alpha = \beta$ and explore its corresponding sample number (the count of high- and low-loss samples) from a predefined set of possibilities. By leveraging the worst environment accuracy, as elaborated in Section 3.2, we identify the optimal candidate that ensures uniform accuracy across all environments.

## 4   Experiments

In this section, we evaluate the effectiveness of our proposed method through comprehensive experiments on multiple datasets and compare it with various methods and baselines. We begin by briefly

---

[1]Note that in the CelebA dataset, only the "blond hair" class includes a minority group.

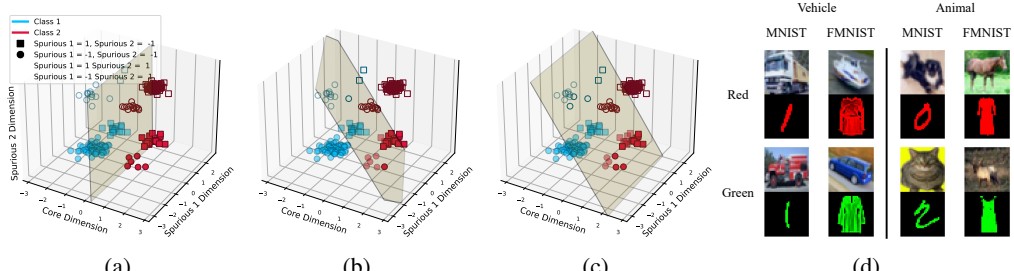

(a)            (b)            (c)            (d)

Figure 3: Two spurious correlations in a dataset. (a) If both spurious attributes are known, they can be utilized to fit a classifier that captures the essential attributes. (b) In the absence of knowledge about both spurious attributes, the model would depend on them for classification, leading to incorrect classification of minority samples. (c) If one spurious attribute is unknown (Spurious 2), the model becomes robust only to the known spurious correlation (Spurious 1), but it still underperforms on minority samples. (d) The Dominoes-CMF dataset, which contains two spurious attributes.

describing evaluation datasets and then introduce baselines and comparative methods. Finally, we report and fully explain the results.

**Datasets**  Our method, along with other baselines, is evaluated on Waterbirds [7], CelebA [13], UrbanCars [14], CivilComments [16], and MultiNLI [15]. As per the study by Yang et al. [4], Waterbirds, CelebA, and UrbanCars among these datasets exhibit spurious correlation. Among the rest, CivilComments has class and attribute imbalance, whereas MultiNLI exhibits attribute imbalance. For additional details on the datasets, please refer to the Appendix.

**Baselines**  We compare our method with four baselines in addition to standard ERM. **GroupDRO** [7] trains a model on the data with the objective of minimizing its average loss on the minority samples. This method requires group labels of both the training and validation sets. **DFR** [1] argues that models trained with ERM are capable of extracting the core features of images. Thus, it first trains a model with ERM, and retrains only the last linear classifier layer on a group-balanced subset of the validation or the held-out training data. While DFR reduces the number of group-annotated samples, it still requires group labels in the training phase. **GroupDRO + EIIL** [12] infers environments of the training set and trains a model with GroupDRO on the inferred environments. **JTT** [5] first trains a model with ERM on the dataset, and then retrains it on the dataset by upweighting the samples that were misclassified by the initial ERM model. **AFR** [9] trains a model with ERM on a portion of the training set, and retrains the classifier on the weighted held-out training data. The weights assigned to retraining samples are based on the loss of the ERM model, upweighting samples from the minority groups. Group DRO + EIIL, JTT and AFR remove the reliance on group annotation in the training phase. However, unlike our method, they all require group labels for model selection.

**Setup**  Similar to all the works mentioned in Section 4, we use ResNet-50 [25] pretrained on ImageNet [26] for image classification tasks. We used random crop and random horizontal flip as data augmentation, similar to [1]. For a fair comparison with the baselines, we did not employ any data augmentation techniques in the process of retraining the last layer of the model. For the CivilComments and MultiNLI, we use pretrained BERT [27] and crop sentences to 220 tokens length. In EvaLS, we use the implementation of EIIL by `spuco` package [28] for environments inference on the model selection set with 20000 steps, SGD optimizer, and learning rate $10^{-2}$ for all datasets.

Model selection and hyper-parameter fine-tuning are done according to the worst environment(or group if annotations are assumed to be available) accuracy on the validation set. For each dataset, we assess the performance of our model in two cases: fine-tuning the ERM classifier or retraining it. For all datasets except MultiNLI, retraining yielded better validation results. We report the results of our experiments in two settings: (i) EVaLS, which incorporates loss-based instance sampling for training the last layer, and environment inference for model selection. (ii) EVaLS-GL, similar to EVaLS except in using ground-truth group labels for model selection. For more details on the ERM training and last layer re-training hyperparameters refer to the Appendix.

## 4.1 Results

The results of our experiments along with the reported results on GroupDRO [7], DFR [1], JTT [5], and AFR [9] on five datasets are shown in Table 1. The reported results for GroupDRO, DFR, JTT, and AFR except those for the UrbanCars are taken from Qiu et al. [9]. For EIIL+Group DRO, the results are reported from Zhang et al. [24]. We report only the worst group accuracy of methods in Table 1. The average group accuracies are documented in the Appendix. The Group Info column shows whether group annotation is required for training or model selection entry for each method.

When compared to other methods with the same level of supervision, EVaLS-GL outperforms on four of the five datasets, achieving near-optimal worst group accuracy on Waterbirds, demonstrating the effectiveness of loss-based sample selection compared to the weighting scheme in AFR [9]. Given that AFR employs exponential weights with a temperature parameter to assign a positive weight to all samples, proportional to the model's assigned probability of the correct class, an increase in the number of low-loss samples will lead to a corresponding rise in their cumulative weight. Consequently, in situations where spurious correlation is high and an uptick in majority samples leads to a greater proportion of low-loss over high-loss samples, determining an appropriate parameter becomes challenging.

The comparison between EVaLS and Group DRO + EIIL indicates that when environments are available instead of groups, our method, which uses environments solely for model selection and utilizes loss-based sampling, is more effective than GroupDRO, a potent invariant learning method, which uses this annotation for training.

Regarding the UrbanCars, which contains an un-annotated spurious attribute, Li et al. [14] has shown that shortcut mitigation methods often struggle to address multiple shortcuts simultaneously. Notably, techniques such as DFR [1] which are designed to reduce reliance on a specific shortcut feature, cannot make the model robust to an unknown shortcut. In contrast, our experiments suggest that loss-based methods can mitigate the impact of both labeled and unlabeled shortcut features more effectively. Also, in the case of CivilComments, which is viewed as a benchmark for class imbalance, EVaLS-GL exceeds all prior methods, even those with complete group annotation, thanks to the class balancing for the training of the last layer.

Our evaluation of EVaLS is based on the spurious correlation benchmarks. This is because, in other instances of subpopulation shift, the attributes that differ across groups are not predictive of the label, thereby reducing the visibility of these attributes' effects in the model's final layers [29]. Consequently, EIIL, which depends on output logits for prediction, might not effectively separate the groups. This observation is further supported by our findings related to the degree of group shift between the environments inferred by EIIL for each class in the CivilComments and MultiNLI datasets. The average group shift (defined in the Section 3.2) in the environments of the minority class of CivilComments is only $5.6_{\pm 0.8}\%$. Also, environments associated with Classes 1 and 2 in MultiNLI show only $1.1_{\pm 0.3}\%$ and $1.9_{\pm 1.0}\%$ group shift respectively. More results and ablation studies can be found in the Appendix.

**Mitigating Multiple Shortcut Attributes**   To evaluate the performance of our method in the case of unknown spurious correlations, we train a ResNet-18 [25] model on the *Dominoes-CMF* dataset. We apply DFR [1], EVaLS-GL, and EVaLS on top of the trained ERMs to assess their ability to mitigate multiple shortcuts. For the last layer training set, we consider the MNIST/Fashion-MNIST feature as the known group label, and the color as the unknown attribute. The results are shown in Table 2. To clarify, we calculate the worst-group accuracy on the validation set considering only the label of one shortcut, *i.e.*, the lowest accuracy among the four groups based on the combination of the target label and the single known shortcut label. Note that EVaLS does not require group annotations.

Our results confirm findings by Li et al. [14], suggesting that methods using group labels mitigate reliance on the known shortcut but not necessarily on the unknown one. EVaLS-GL mitigates this phenomenon using its loss-based sampling approach, but surprisingly EVaLS even outperforms EVaLS-GL. Combining a loss-based sampling approach for last layer training and environment-based model selection, results in a completely group-annotation-free method in a multi-shortcut setting and successfully re-weights features to perform well with respect to both spurious attributes.

Table 1: A comparison of the worst group accuracy across various methods, ours included, on five datasets. The Group Info column indicates if each method utilizes group labels of the training/validation data, with ✓✓ denoting that group information is employed during both the training and validation stages. Bold numbers are the highest results overall, while underlined ones are the best among methods that may require group annotation only for model selection. CivilComments is class imbalanced, MultiNLI has imbalanced attributes, and the other three datasets have spurious correlations. The × sign indicates that the dataset is out of the scope of the method. The mean and standard deviation are calculated over three runs with different seeds.

| Method | Group Info | | Datasets | | | | |
| | Train/Val | Waterbirds | CelebA | UrbanCars | CivilComments | MultiNLI |
|---|---|---|---|---|---|---|
| GDRO [7] | ✓/✓ | 91.4 | **88.9** | - | 69.9 | **77.7** |
| DFR [1] | ✗/✓✓ | **92.9**$_{\pm 0.2}$ | 88.3$_{\pm 1.1}$ | 79.6$_{\pm 2.22}$ | 70.1$_{\pm 0.8}$ | 74.7$_{\pm 0.7}$ |
| GDRO + EIIL [12] | ✗/✓ | 77.2$_{\pm 1}$ | 81.7$_{\pm 0.8}$ | - | 67.0$_{\pm 2.4}$ | - |
| JTT [5] | ✗/✓ | 86.7 | 81.1 | - | 69.3 | 72.6 |
| AFR [9] | ✗/✓ | $\underline{90.4}_{\pm 1.1}$ | 82.0$_{\pm 0.5}$ | 80.2$_{\pm 2.0}$ | 68.7$_{\pm 0.6}$ | $\underline{73.4}_{\pm 0.6}$ |
| EVaLS-GL (Ours) | ✗/✓ | 89.4$_{\pm 0.3}$ | $\underline{84.6}_{\pm 1.6}$ | **82.27**$_{\pm 1.16}$ | **80.5**$_{\pm 0.4}$ | $\underline{75.1}_{\pm 1.2}$ |
| ERM | ✗/✗ | 66.4$_{\pm 2.3}$ | 47.4$_{\pm 2.3}$ | 18.67$_{\pm 2.01}$ | 61.2$_{\pm 3.6}$ | 64.8$_{\pm 1.9}$ |
| EVaLS (Ours) | ✗/✗ | 88.4$_{\pm 3.1}$ | $\underline{85.3}_{\pm 0.4}$ | 82.13$_{\pm 0.92}$ | × | × |

Table 2: Worst test group accuracy of ERM, DFR, EVaLS, and EVaLS-GL on the Dominoes-CMF Dataset. The mean and standard deviation are calculated based on runs with three distinct seeds.

| | ERM | DFR | EVaLS-GL | EVaLS |
|---|---|---|---|---|
| Worst Group Accuracy | 50.6$_{\pm 1.0}$ | 60.2$_{\pm 1.2}$ | 63.6$_{\pm 1.3}$ | **67.1**$_{\pm 4.2}$ |

## 5 Discussion

This study presents EVaLS, a novel approach to improve robustness to spurious correlations with zero group annotation. EVaLS uses loss-based sampling to create a balanced training dataset that effectively disrupts spurious correlations and employs EIIL to infer environments for model selection. We also explore situations with multiple spurious correlations where not all spurious factors are known. In this context, we introduce Dominoes-CMF, a dataset in which two factors are spuriously correlated with the label, but only one is identified. Our findings suggest that EVaLS attains near-optimal worst test group accuracy on spurious correlation datasets. We also present EVaLS-GL, which needs group labels only for model selection. Our empirical tests on various datasets demonstrate EVaLS-GL outperforms state-of-the-art methods requiring group data during evaluation or training.

Note that this paper remains consistent with the findings of Lin et al. [30]. Our approach does not involve identifying spurious attributes without auxiliary information. Instead, the objective is to make a trained model robust against its reliance on shortcuts. Specifically, conditioning on what a trained model learns, we ascertain that both the loss value and the model's feature space are instrumental in mitigating shortcuts and effectuating notable shifts among groups.

EVaLS and EVaLS-GL may struggle with small datasets due to a low number of selected samples for the last layer training. Also, as environment inference from the last layer features is not effective for all types of subpopulation shifts, EVaLS is limited to datasets with spurious correlation. Similar to other methods in the field, EVaLS prioritizes the worst group accuracy at the cost of less average accuracy. Additionally, a notable variance has been observed in some of our experiments.

EVaLS represents a significant advancement in the development of methods for enhancing model fairness and robustness without prior knowledge about group annotations. Future work could explore developing environment inference methods effective for other types of subpopulation shift, such as attribute and class imbalance.

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

# A   Related Work

Robustness to spurious correlation is a critical concern across various machine learning subfields. It is a form of out-of-distribution generalization [31] where the distribution shift arises from the disproportionate representation of minority groups—those instances that are devoid of the correlated spurious patterns associated with their labels [4]. The issue of spurious correlation also intersects with the discourse on fairness in machine learning. [32, 33].

Past studies have proposed a range of strategies to mitigate the models' reliance on spurious correlation. Broadly speaking, these methods can be categorized according to the degree of supervision they require regarding group labels.

Invariant learning (IL) methods  [18, 34, 35] operate under the assumption of having access to a collection of environments that comprise group shift. By imposing invariant conditions on these environments, IL methods strive to create classifiers robust against group-sensitive features. IRM [18] is designed to learn a feature extractor, which, when utilized, guarantees the existence of a classifier that would be optimal in all training environments. VREx [34] aims to decrease the risk variance among different training environments. PGI [36] works by minimizing the distance between the expected softmax distribution of labels, conditioned on inputs across both majority and minority environments. Lastly, Fishr [35] focuses on bringing the variance of risk gradients closer together across different training environments. For scenarios which environments are not available, environment inference methods [12, 22] are used to obtain a set of environments. Creager et al. [12] introduce environment inference for invariant learning (EIIL), which tries to partition samples into two groups such that the objective of IRM [18] is maximized. HRM [22] aims to optimize both an environment inference module and an invariant prediction module jointly, with the goal of achieving an invariant predictor.

When group annotations are accessible, various methods leverage this information to equalize the impact of different groups on the model's loss. The Group Distributionally Robust Optimization (GDRO) approach [7], for instance, focuses on optimizing the loss for the worst-performing group during training. Kirichenko et al. [1] has shown that models can still learn and extract core data features even in the presence high spurious correlation. Consequently, They suggest that retraining just the last layer of a model initially trained with Empirical Risk Minimization (ERM) can effectively reduce reliance on spurious correlation for predicting class labels. This method, termed Deep Feature Re-weighting (DFR), has been validated as not only highly effective but also significantly more efficient than earlier techniques that necessitated retraining the full model [8, 7]. However, availability of group annotations is considered a serious restrictive assumption.

Several recent studies have endeavored to enhance model robustness against spurious correlation, even in the absence of group annotations [5, 24, 9, 2, 6]. Liu et al. [5] introduce a two-stage method that involves training a model using ERM for a number of epochs before retraining it to give more weight to misclassified samples. The study by Zhang et al. [24] employs the same two-stage training process, but with a twist for the second stage: they utilize contrastive methods. The goal is to bring samples from the same class but with divergent predictions closer in the feature space, while simultaneously increasing the separation between samples from different classes that have similar predictions. Another method, known as automatic feature reweighting (AFR) [9], reweights the last layer of an ERM-pretrained model to favor samples that the original model was less accurate on. LaBonte et al. [2] refine the last layer of an ERM-trained model through class-balanced finetuning, identifying challenging data points by comparing the classifier's predictions with those of an early-stopped version. While these methods have significantly reduced the reliance on group annotations, some are still required for validation and model selection. This remains a constraint, particularly when the spurious correlation is completely unknown.

For making a trained model robust to spurious correlation with zero group annotations, recently, LaBonte et al. [2] have empirically demonstrated that the class-balanced retraining of a model pretrained with ERM can effectively improve the WGA for certain datasets. However, this approach fails in datasets with a high degree of spurious correlation.

Table 3: The average and variation percentage (%)(across 3 seeds) of group shift between the inferred environments using EIIL [12] for each class, which is the absolute difference between the proportion of a minority group in the two environments of a class. Higher group shift indicates better separation of environments. In most cases, a significant group shift is observed between the inferred environments.

| Class No. | Dataset | | |
| --- | --- | --- | --- |
| | Waterbirds | CelebA | UrbanCars |
| 0 | $16.6_{\pm 0.7}$ | $3.6_{\pm 0.2}$ | $17.7_{\pm 1.2}, 23.5_{\pm 0.1}, 62.1_{\pm 1.9}$ |
| 1 | $50.5_{\pm 0.3}$ | $14.1_{\pm 0.9}$ | $40.7_{\pm 7.9}, 13.8_{\pm 0.1}, 19.2_{\pm 3.9}$ |

## B  Environment Inference for Invariant Learning

Consider the training dataset $\mathcal{D}^{tr} = \{(x^{(i)}, y^{(i)}) | x^{(i)} \in \mathcal{X}, y^{(i)} \in \mathcal{Y}\}$, where $\mathcal{X}$ and $\mathcal{Y}$ represent the input and output spaces, respectively. This dataset can be partitioned into different environments $\mathcal{E}^{tr} = \{e_1, ..., e_n\}$, such that for any $i \neq j$, the data distribution in $e_i$ and $e_j$ differs. The objective of invariant learning is to train a predictor that performs consistently across all environments in $\mathcal{E}^{tr}$. Under certain conditions, this predictor is also expected to perform well on $e^{tst}$, a test environment with a distribution distinct from the training data. Invariant Risk Minimization (IRM) [18] approaches this problem by learning a feature extractor $\Phi(.)$ such that a classifier $\omega(.)$ exists, where $\omega \circ \Phi(.)$ performs consistently across all training environments. The practical implementation of the IRM objective is to minimize

$$\sum_{e \in \mathcal{E}^{tr}} R^e(\Phi) + \lambda ||\nabla_{\bar{\omega}} R^e(\bar{\omega} \circ \Phi)||^2, \tag{2}$$

where $\bar{\omega}$ is a constant scalar with a value of 1.0, $\lambda$ is a hyperparameter, and $R^e(f) = \mathbf{E}_{(x,y) \sim p_e}[l(f(x), y)]$ is referred to as the risk on environment $e$.

In real-world scenarios, training environments might not always be available. To address this, Environment Inference for Invariant Learning (EIIL) [12] partitions samples into two environments in a way that maximizes the objective in Eq 2.

During the training phase, the EIIL algorithm replaces the hard assignment of environments to samples with a soft assignment $\mathbf{q}_i(e) = p(e|(x^{(i)}, y^{(i)}))$, where $\mathbf{q}_i$ is learnable. Consequently, the relaxed version of the risk function is defined as $\tilde{R}^e(\Phi) = \frac{1}{N} \sum_i^N \mathbf{q}_i(e)[l(\Phi(x^{(i)}), y^{(i)})]$. Given a model $\Phi$ that has been trained with ERM on the dataset, EIIL optimizes

$$\mathbf{q}^* = \arg\max_{\mathbf{q}} ||\nabla_{\bar{\omega}} \tilde{R}^e(\bar{\omega} \circ \Phi)||. \tag{3}$$

As discussed in Creager et al. [12], using a biased base model $\Phi$ could lead to environments exhibiting varying degrees of spurious correlation. During the inference phase, the soft assignment is converted to a hard assignment. The average group shift between the inferred environments using EIIL is illustrated in Table 3.

## C Theoretical Analysis

In this section, we establish a more formal description of loss-based sampling for balanced dataset creation and then prove it. We thoroughly analyze the close relationship between the availability of the balanced dataset and the gap between spurious features of minority and majority groups.

Consider a binary classification problem with a cross-entropy loss function. Let logits be denoted as $L$. Because loss is a monotonic function of logits, the tails of the distribution of loss across samples are equivalent to that of the logits in each class. We assume that in feature space (output of $g_\theta$) samples from the minority and majority of a class are derived from Gaussian distributions $\mathcal{N}(h_{\min}, t_{\min}^2 I_d)$ and $\mathcal{N}(h_{\text{maj}}, t_{\text{maj}}^2 I_d)$, respectively. Before diving into the group balance problem we initially show that the distribution of minority and majority samples in the logit space (output of $h_\phi$) are Gaussian too.

**Lemma C.1** (Gaussain Distribution of Logits). *if $Z \sim \mathcal{N}(h, t^2 I_d)$ in feature space and $W \in \mathbb{R}^d$ then logits $L = \langle W, Z \rangle \sim \mathcal{N}(Wh,\, t^2 \|W\|^2)$*

*Proof.* Let $Z \sim \mathcal{N}(h, t^2 I_d)$.

Consider the linear combination $L = \langle W, Z \rangle = W^T Z$, where $W \in \mathbb{R}^d$ which is a univariate gaussian.

To find the distribution of $L$, we need to determine its mean and variance.

1. **Mean of $L$**

$$\mathbb{E}[L] = \mathbb{E}[\langle W, Z \rangle] = \mathbb{E}[W^T Z] = W^T \mathbb{E}[Z] = W^T h = \langle W, h \rangle.$$

Therefore, the mean of $L$ is $Wh$.

2. **Variance of $L$**:

The variance of $L$ can be computed using the properties of covariance. Recall that if $Z \sim \mathcal{N}(h, t^2 I_d)$, then the covariance matrix of $Z$ is $t^2 I_d$.

The variance of the linear combination $L = W^T Z$ is given by:

$$\text{Var}(L) = \text{Var}(W^T Z) = W^T \text{Cov}(Z) W.$$

Given $\text{Cov}(Z) = t^2 I_d$, we have:

$$\text{Var}(L) = W^T (t^2 I_d) W = t^2 W^T I_d W = t^2 \|W\|^2,$$

where $\|W\|$ denotes the Euclidean norm of $W$.

Combining the mean and variance results, we conclude that $L$ is normally distributed with mean $Wh$ and variance $t^2 \|W\|^2$:

$$L = \langle W, Z \rangle \sim \mathcal{N}(Wh, t^2 \|W\|^2).$$

Thus, we have proved that if $Z \sim \mathcal{N}(h, t^2 I_d)$, then the logits $L = \langle W, Z \rangle$ follow the distribution $\mathcal{N}(Wh, t^2 \|W\|^2)$. $\square$

From now on, we consider $\mathcal{N}(\mu_{\min}, \sigma_{\min}^2)$ and $\mathcal{N}(\mu_{\text{maj}}, \sigma_{\text{maj}}^2)$ as the distribution of minority and majority samples in logits space.

Next, we prove the more formal version of the main proposition 3.1 which describes the existence of a balanced dataset, only after we define a key concept, *proportional density difference* (illustrated in figure 4) to outline our proof.

**Definition C.1** (Proportional Density Difference). *For any interval $I = (a, b]$ and a mixture distribution $\varepsilon P_1(x) + (1 - \varepsilon)P_2(x)$, proportional density difference is defined by the difference of accumulation of two component distributions in the interval $I$ and is denoted by $\Delta_\varepsilon P_{mixture}(I)$.*

$$\Delta_\varepsilon P_{mixture}(I) \triangleq \varepsilon P_1\big(x \in I\big) - (1 - \varepsilon)P_2\big(x \in I\big)$$

**Definition C.2** (Tail Proportional Density Difference). *For a mixture distribution $\varepsilon P_1(x) + (1 - \varepsilon)P_2(x)$, we define $tail_L(\alpha)$ as $\Delta_\varepsilon P_{mixture}\big((-\infty, \alpha]\big)$ and $tail_R(\beta)$ as $-\Delta_\varepsilon P_{mixture}\big((\beta, +\infty)\big)$.*

**Corollary C.1.**

$$tail_L(\alpha) = \varepsilon F^1(\alpha) - (1 - \varepsilon)F^2(\beta)$$
$$tail_R(\alpha) = (1 - \varepsilon)\big[1 - F^2(\beta)\big] - \varepsilon\big[1 - F^1(\beta)\big]$$

*where $F^1$ and $F^2$ are CDF of two component distributions.*

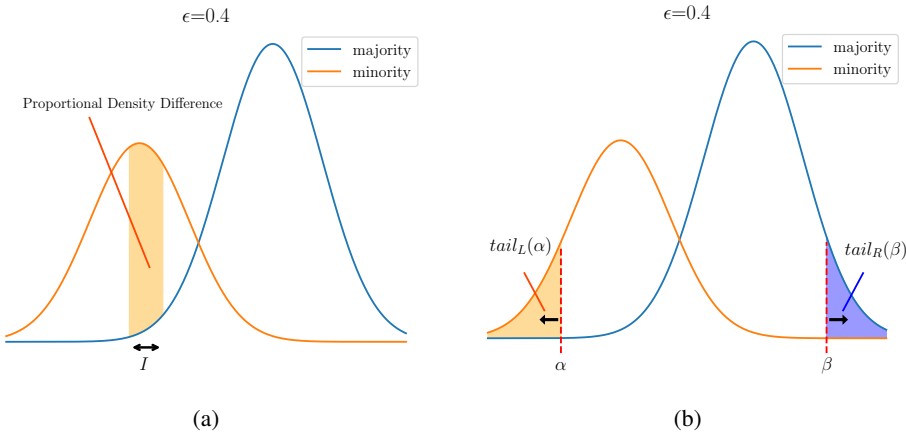

(a)           (b)

Figure 4: (a) Illustration of proportion density difference C.1, (b) equation of $tail_L(\alpha) = tail_R(\beta)$ at C.2.

**Proposition C.1** (Feasiblity Of Loss-based Group Balancing). *Suppose that $L$ is derived from the mixture of two distributions $\mathcal{N}(\mu_{min}, \sigma_{min}^2)$ and $\mathcal{N}(\mu_{maj}, \sigma_{maj}^2)$ with proportion of $\varepsilon$ and $1 - \varepsilon$, respectively, where $\varepsilon \leq \frac{1}{2}$. There exists $\alpha$ and $\beta$ such that restricting $L$ to the $\alpha$-left and $\beta$-right tails of its distribution results in a group-balanced distribution if and only if $\sigma_{min} \geq \sigma_{maj}$ or*

$$tail_L(\frac{-B + \sqrt{\Delta}}{2A}) > 0 \tag{4}$$

*and*

$$\epsilon \geq sigmoid\left(-\frac{(\mu_{maj} - \mu_{min})^2}{2(\sigma_{maj}^2 - \sigma_{min}^2)} - \log\left(\frac{\sigma_{maj}}{\sigma_{min}}\right)\right) \tag{5}$$

*where $A = \left(\frac{1}{2\sigma_{maj}^2} - \frac{1}{2\sigma_{min}^2}\right)$, $B = \left(\frac{\mu_{min}}{\sigma_{min}^2} - \frac{\mu_{maj}}{\sigma_{maj}^2}\right)$ and $\Delta = \frac{(\mu_{min} - \mu_{maj})^2}{\sigma_{min}^2 \sigma_{maj}^2} - 4\left[\log\left(\frac{\sigma_{maj}}{\sigma_{min}}\right) + \log\left(\frac{\epsilon}{1-\epsilon}\right)\right]\left[\frac{1}{2\sigma_{maj}^2} - \frac{1}{2\sigma_{min}^2}\right]$.*

**Proof outline** Our proof proceeds with three steps. First, we reformulate the theorem as an equality of left- and right-tail proportional distribution differences. In other words, we show that the more mass the minority distribution has on one tail, the more mass the majority distribution must have on the other tail. Afterward, supposing $\mu_{min} < \mu_{maj}$ WOLG, we propose a proper range for $\beta$ values on the right tail. We show that when $\sigma_{maj} \leq \sigma_{min}$, values for $\alpha$ trivially exist that can overcome the imbalance between the two distributions. In the last step, for the case in which the variance of the majority is higher than the minority, we discuss a necessary and sufficient condition for the existence of $\alpha$ and $\beta$ based on the left-tail proportional density difference using the properties of its derivative with respect to $\alpha$.

**Step 1** *Reformulating the problem based on proportional distribution difference.*

We introduce a utility random variable *Logit Value Tier* as $T$, which is defined as a function of a random variable $L$.

$$T_{\alpha,\beta} = \begin{cases} High & \text{if } L \geq \beta \\ Mid & \text{if } \alpha < L < \beta \\ Low & \text{if } L \leq \alpha \end{cases} \tag{6}$$

We can rewrite the problem in formal form as finding an $\alpha$ and $\beta$ which satisfies the following equation:

$$P\Big(g = \min\Big|T_{\alpha,\beta} \neq Mid\Big) = P\Big(g = \text{maj}\Big|T_{\alpha,\beta} \neq Mid\Big) \tag{7}$$

Equation 5 now can be rewritten to a more suitable form:

$$P\Big(g = \min\Big|T_{\alpha,\beta} \neq Mid\Big) = P\Big(g = \text{maj}\Big|T_{\alpha,\beta} \neq Mid\Big) \tag{8}$$

$$\Longleftrightarrow \quad \frac{P\Big(T_{\alpha,\beta} \neq Mid\Big|g = \min\Big)P(g = \min)}{P\Big(T_{\alpha,\beta} \neq Mid\Big)} = \frac{P\Big(T_{\alpha,\beta} \neq Mid|g = \text{maj}\Big)P(g = \text{maj})}{P\Big(T_{\alpha,\beta} \neq Mid\Big)} \tag{9}$$

$$\Longleftrightarrow \quad P\Big(T_{\alpha,\beta} \neq Mid\Big|g = \min\Big)P(g = \min) = P\Big(T_{\alpha,\beta} \neq Mid\Big|g = \text{maj}\Big)P(g = \text{maj}) \tag{10}$$

$$\Longleftrightarrow \quad \varepsilon P\Big(T_{\alpha,\beta} \neq Mid\Big|g = \min\Big) = (1-\varepsilon)P\Big(T_{\alpha,\beta} \neq Mid\Big|g = \text{maj}\Big) \tag{11}$$

$$\Longleftrightarrow \quad \varepsilon\Big[P\Big(T_{\alpha,\beta} = Low\Big|g = \min\Big) + P\Big(T_{\alpha,\beta} = High\Big|g = \min\Big)\Big] = \tag{12}$$

$$(1-\varepsilon)\Big[P\Big(T_{\alpha,\beta} = Low\Big|g = \text{maj}\Big) + P\Big(T_{\alpha,\beta} = High\Big|g = \text{maj}\Big)\Big] \tag{13}$$

$$\Longleftrightarrow \quad \varepsilon\Big[P\Big(L \leq \alpha\Big|g = \min\Big) + P\Big(L \geq \beta\Big|g = \min\Big)\Big] = \tag{14}$$

$$(1-\varepsilon)\Big[P\Big(L \leq \alpha\Big|g = \text{maj}\Big) + P\Big(L \geq \beta\Big|g = \text{maj}\Big)\Big] \tag{15}$$

$$\Longleftrightarrow \quad \varepsilon\Big[F^{\min}(\alpha) + \Big(1 - F^{\min}(\beta)\Big)\Big] = (1-\varepsilon)\Big[F^{\text{maj}}(\alpha) + \Big(1 - F^{\text{maj}}(\beta)\Big)\Big] \tag{16}$$

$$\Longleftrightarrow \quad \varepsilon F^{\min}(\alpha) - (1-\varepsilon)F^{\text{maj}}(\alpha) = (1-\varepsilon)\Big[1 - F^{\text{maj}}(\beta)\Big] - \varepsilon\Big[1 - F^{\min}(\beta)\Big] \tag{17}$$

We can see the left side of equation 17 is just a function of $alpha$. The same goes for the right side of the equation which is a function of $\beta$.

Rewriting the left side of the equation as $tail_L(\alpha)$ and right side as $tail_R(\beta)$, the problem is now reduced to finding an $\alpha$ and $\beta$ that satisfies

$$tail_L(\alpha) = tail_R(\beta) \tag{18}$$

which is shown in figure 4.

Before reaching out to step two we discuss the properties of $tail_L$ and $tail_R$ in Lemma C.2.

**Lemma C.2.** *$tail_L(\alpha)$ and $tail_R(\beta)$ are continuous functions and* $\lim_{\alpha \to -\infty} tail_L(\alpha) = 0$, $\lim_{\alpha \to +\infty} tail_L(\alpha) = 2\varepsilon - 1 < 0$, $\lim_{\beta \to +\infty} tail_R(\beta) = 0$ *and* $\lim_{\beta \to -\infty} tail_R(\beta) = 1 - 2\varepsilon > 0$.

*Proof.* Simply proved by the definition of $tail$ functions and properties of CDF. $\qquad\square$

**Step 2** *Solving the equation 18 for simple cases.*

**Lemma C.3.** $tail_R(\mu_{maj}) > \frac{1}{2} - \varepsilon \geq 0$

*Proof.*

$$tail_R(\mu_{\text{maj}}) = (1-\varepsilon)\Big[1 - F^{\text{maj}}(\mu_{\text{maj}})\Big] - \varepsilon\Big[1 - F^{\text{min}}(\mu_{\text{maj}})\Big] \tag{19}$$

$$= (1-\varepsilon)\Big[1 - \phi(0)\Big] - \varepsilon\Big[1 - \phi\big(\frac{\mu_{\text{maj}} - \mu_{\text{min}}}{\sigma_{\text{min}}}\big)\Big] \tag{20}$$

$$> \frac{(1-\varepsilon)}{2} - \varepsilon\big(1 - \frac{1}{2}\big) = \frac{1 - 2\varepsilon}{2} = \frac{1}{2} - \varepsilon \tag{21}$$

$\square$

**Corollary C.2.** *Because $tail_R$ is continuous and $\lim_{\beta \to +\infty} tail_R(\beta) = 0$, based on the mean value theorem, any value between zero and $\frac{(1-2\varepsilon)}{2}$ is obtainable by selecting a $\beta$ in $[\mu_2, +\infty)$.*

According to the previous corollary C.2 finding a positive $tail_L(\alpha)$ will satisfy our need. to find a suitable point, we employ derivatives and properties of relative PDFs to maximize $tail_L(\alpha)$ and find a positive value.

$$\frac{\mathrm{d}tail_L(\alpha)}{\mathrm{d}\alpha} = \varepsilon f^{\text{min}}(\alpha) - (1-\varepsilon)f^{\text{maj}}(\alpha) = \varepsilon f^{\text{maj}}(\alpha)\Big[\frac{f^{\text{min}}(\alpha)}{f^{\text{maj}}(\alpha)} - \frac{1-\varepsilon}{\varepsilon}\Big] \tag{22}$$

The term $\big[\frac{f^{\text{min}}(\alpha)}{f^{\text{maj}}(\alpha)} - \frac{1-\varepsilon}{\varepsilon}\big]$ has the same sign with derivative of $tail_L(\alpha)$, also it's roots are critical points of $tail_L$, analyzing characteristics of $\log \frac{f^{\text{min}}(\alpha)}{f^{\text{maj}}(\alpha)}$ is the key insight to find a proper $\alpha$ value.

$$\log f^{\text{min}}(\alpha) - \log f^{\text{maj}}(\alpha) = \log\Big(\frac{1-\epsilon}{\epsilon}\Big)$$

$$\Rightarrow \log\Big(\frac{\sigma_{\text{maj}}}{\sigma_{\text{min}}}\Big) - \log\Big(\frac{1-\epsilon}{\epsilon}\Big) - \frac{(\alpha - \mu_{\text{min}})^2}{2\sigma_{\text{min}}^2} + \frac{(\alpha - \mu_{\text{maj}})^2}{2\sigma_{\text{maj}}^2} = 0$$

$$\Rightarrow \Big(\frac{1}{2\sigma_{\text{maj}}^2} - \frac{1}{2\sigma_{\text{min}}^2}\Big)\alpha^2 + \Big(\frac{\mu_{\text{min}}}{\sigma_{\text{min}}^2} - \frac{\mu_{\text{maj}}}{\sigma_{\text{maj}}^2}\Big)\alpha + \Big[\frac{\mu_{\text{maj}}^2}{2\sigma_{\text{maj}}^2} - \frac{\mu_{\text{min}}^2}{2\sigma_{\text{min}}^2} + \log\Big(\frac{\sigma_{\text{maj}}}{\sigma_{\text{min}}}\Big) + \log\Big(\frac{\epsilon}{1-\epsilon}\Big)\Big] = 0$$

Because $\lim_{\alpha \to -\infty} tail_L(\alpha) = 0$ and $\lim_{\beta \to +\infty} tail_R(\beta) < 0$ to have a positive $tail_L(\alpha)$ we need to have an interval which $\frac{\mathrm{d}tail_L(\alpha)}{\mathrm{d}\alpha}$ is positive, for a second degree polynomial like $ax^2 + bx + c$ to have positive value, either $a \geq 0$ or $\Delta > 0$, in our case $a$ is $\big(\frac{1}{\sigma_{\text{maj}}^2} - \frac{1}{\sigma_{\text{min}}^2}\big)$. if $\sigma_{\text{min}} \geq \sigma_{\text{maj}}$ then $a \geq 0$ and the minority CDF function will dominate the majority CDF function in the left-side tail and by choosing a negative number with big enough absolute value for alpha and $tail_L(\alpha)$ will be positive.

**Step 3** *Solving equation 18 for special case $\sigma_{min} < \sigma_{maj}$* In case of $\sigma_{\text{min}} \leq \sigma_{\text{maj}}$, having $\Delta > 0$ is a necessary condition, also derivative of $tail_L(\alpha)$ is only positive in $(\frac{-b-\sqrt{\Delta}}{2a}, \frac{-b+\sqrt{\Delta}}{2a})$ so the maximum of $tail_L$ is either in $-\infty$ or in $\frac{-b+\sqrt{\Delta}}{2a}$. Having $tail_L(\frac{-b+\sqrt{\Delta}}{2a}) > 0$ next to $\Delta > 0$ condition, would be the necessary and also sufficient in this case.

$$B^2 = \frac{\mu_{\text{min}}^2}{\sigma_{\text{min}}^4} + \frac{\mu_{\text{maj}}^2}{\sigma_{\text{maj}}^4} - 2\frac{\mu_{\text{maj}}\mu_{\text{min}}}{\sigma_{\text{maj}}^2\sigma_{\text{min}}^2}$$

$$4AC = \frac{\mu_{\text{min}}^2}{\sigma_{\text{min}}^4} - \frac{\mu_{\text{min}}^2}{\sigma_{\text{maj}}^2\sigma_{\text{min}}^2} - \frac{\mu_{\text{maj}}^2}{\sigma_{\text{maj}}^2\sigma_{\text{min}}^2} + \frac{\mu_{\text{maj}}^2}{\sigma_{\text{maj}}^4} + 4\Big[\log\Big(\frac{\sigma_{\text{maj}}}{\sigma_{\text{min}}}\Big) + \log\Big(\frac{\epsilon}{1-\epsilon}\Big)\Big]\Big[\frac{1}{2\sigma_{\text{maj}}^2} - \frac{1}{2\sigma_{\text{min}}^2}\Big]$$

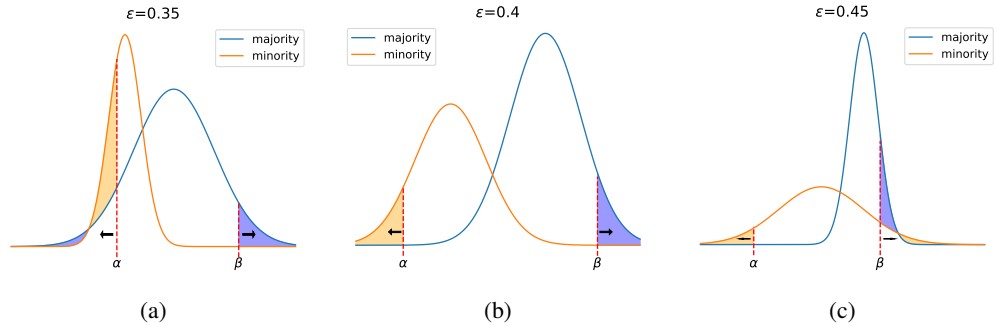

Figure 5: Tail thresholds for three cases: (a) minority group variance is less than majority ($\sigma_{\text{min}} < \sigma_{\text{maj}}$), (b) the variance of two groups are equal ($\sigma_{\text{min}} = \sigma_{\text{maj}}$) and (c) the variance of the minority group is more than majority ($\sigma_{\text{min}} > \sigma_{\text{maj}}$).

$$\Delta = \frac{(\mu_{\text{min}} - \mu_{\text{maj}})^2}{\sigma_{\text{min}}^2 \sigma_{\text{maj}}^2} - 4\left[\log\left(\frac{\sigma_{\text{maj}}}{\sigma_{\text{min}}}\right) + \log\left(\frac{\epsilon}{1-\epsilon}\right)\right]\left[\frac{1}{2\sigma_{\text{maj}}^2} - \frac{1}{2\sigma_{\text{min}}^2}\right] \geq 0$$

$$\iff (\mu_{\text{min}} - \mu_{\text{maj}})^2 \geq 2\left[\log\left(\frac{1-\epsilon}{\epsilon}\right) - \log\left(\frac{\sigma_{\text{maj}}}{\sigma_{\text{min}}}\right)\right]\left[\sigma_{\text{maj}}^2 - \sigma_{\text{min}}^2\right]$$

$$\iff \epsilon \geq \text{sigmoid}\left(-\frac{(\mu_{\text{maj}} - \mu_{\text{min}})^2}{2(\sigma_{\text{maj}}^2 - \sigma_{\text{min}}^2)} - \log\left(\frac{\sigma_{\text{maj}}}{\sigma_{\text{min}}}\right)\right)$$

Next, we investigate properties of the conditions of the proposition C.1 in case of $\sigma_{\text{maj}} < \sigma_{\text{min}}$. Schematic interpretation of these conditions is presented in figure 6.

- As equation 5 indicates, the minority group is not allowed to be too underrepresented. This especially has a direct relation with the difference of means. The more mean values of groups are different, the more imbalance can be mitigated through loss-based sampling. Mean value difference is especially affected by the spurious correlation, it escalates as the model relies on spurious correlation and also when the spurious features between groups are too different.

- On the other hand condition 4 is more complex and doesn't have a simple closed form, we analytically describe its behaviors by fixating the means and calculating the valid values for $\varepsilon$. As the results show in figure 6, most of $\varepsilon$ are feasible in for $\sigma_{\text{min}} < \Delta\mu$ as we can see the possible region declines with an increase of $\sigma_{\text{min}}$ and valid $\varepsilon$ values cease to exist.

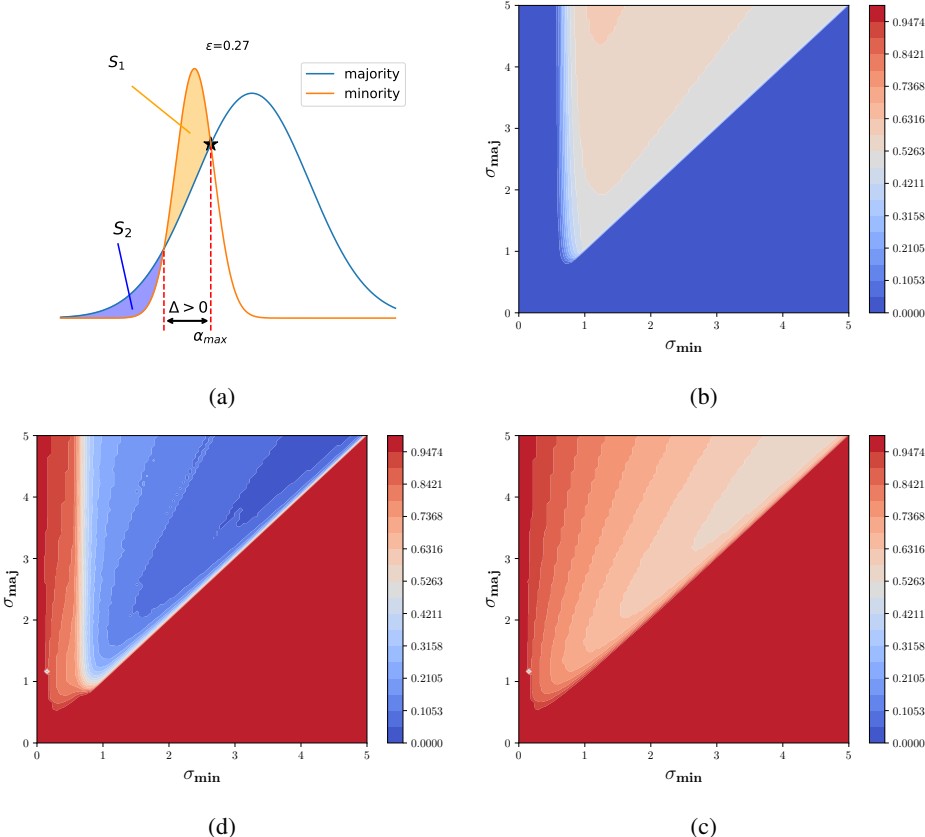

Figure 6: (a) Conditions if $\sigma_{\min} > \sigma_{\text{maj}}$, (b), (c), (d) Minimum, maximum and interval length of feasible $\varepsilon$ values across $(\sigma_{\min}, \sigma_{\text{maj}})$ field for $\mu_{\min} = 0$, $\mu_{\text{maj}} = 1$.

Table 4: A comparison of the various methods, ours included, on spurious correlation datasets. The Group Info column indicates if each method utilizes group labels of the training/validation data, with ✓ denoting that group information is employed during both the training and validation stages. Both the average test accuracy and worst test group accuracy are reported. The mean and standard deviation are calculated over three runs with different seeds. The numbers in bold represent the highest results among all methods, while the underlined numbers represent the best results among methods that do not require group annotation in the training phase.

| Method | Group Info | Waterbirds | | CelebA | | UrbanCars | |
| --- | --- | --- | --- | --- | --- | --- | --- |
| | Train/Val | Worst | Best | Worst | Best | Worst | Best |
| GDRO [7] | ✓/✓ | 91.4 | 93.5 | **88.9** | 92.9 | - | - |
| DFR [1] | ✗/✓✓ | **92.9**$_{\pm0.2}$ | 94.2$_{\pm0.4}$ | 88.3$_{\pm1.1}$ | 91.3$_{\pm0.3}$ | 79.6$_{\pm2.22}$ | 87.5$_{\pm0.6}$ |
| GDRO + EIIL [12] | ✗/✓ | 77.2$_{\pm1}$ | 96.5$_{\pm0.2}$ | 81.7$_{\pm0.8}$ | 85.7$_{\pm0.1}$ | - | - |
| JTT [5] | ✗/✓ | 86.7 | 93.3 | 81.1 | 88.0 | - | - |
| AFR [9] | ✗/✓ | 90.4$_{\pm1.1}$ | 94.2$_{1.2}$ | 82.0$_{\pm0.5}$ | 91.3$_{\pm0.3}$ | 80.2$_{\pm2.0}$ | 87.1$_{\pm1.2}$ |
| EVaLS-GL (Ours) | ✗/✓ | 89.4$_{\pm0.3}$ | 95.1$_{\pm0.3}$ | 84.6$_{\pm1.6}$ | 91.1$_{\pm0.6}$ | **82.27**$_{\pm1.16}$ | **88.2**$_{\pm0.6}$ |
| ERM | ✗/✗ | 66.4$_{\pm2.3}$ | 90.3$_{\pm0.5}$ | 47.4$_{\pm2.3}$ | **95.5**$_{\pm0.0}$ | 18.67$_{\pm2.01}$ | 76.5$_{\pm4.6}$ |
| EVaLS (Ours) | ✗/✗ | 88.4$_{\pm3.1}$ | 94.1$_{\pm0.1}$ | 85.3$_{\pm0.4}$ | 89.4$_{\pm0.5}$ | 82.13$_{\pm0.92}$ | 88.1$_{\pm0.9}$ |

Table 5: A comparison of the various methods, ours included, on CivilComments and MultiNLI. The Group Info column indicates if each method utilizes group labels of the training/validation data, with ✓ denoting that group information is employed during both the training and validation stages. Both the average test accuracy and worst test group accuracy are reported. The mean and standard deviation are calculated over three runs with different seeds. The numbers in bold represent the highest results among all methods, while the underlined numbers represent the best results among methods that do not require group annotation in the training phase.

| Method | Group Info | CivilComments | | MultiNLI | |
| --- | --- | --- | --- | --- | --- |
| | Train/Val | Worst | Best | Worst | Best |
| GDRO [7] | ✓/✓ | 69.9 | 88.9 | **77.7** | 81.4 |
| DFR [1] | ✗/✓✓ | 70.1$_{\pm0.8}$ | 87.2$_{\pm0.3}$ | 74.7$_{\pm0.7}$ | 82.1$_{\pm0.2}$ |
| GDRO + EIIL [12] | ✗/✓ | 67.0$_{\pm2.4}$ | 90.5$_{\pm0.2}$ | - | - |
| JTT [5] | ✗/✓ | 69.3 | 91.1 | 72.6 | 78.6 |
| AFR [9] | ✗/✓ | 68.7$_{\pm0.6}$ | 89.8$_{\pm0.6}$ | 73.4$_{\pm0.6}$ | 81.4$_{\pm0.2}$ |
| EVaLS-GL (Ours) | ✗/✓ | **80.5±0.4** | 88.0$_{\pm0.4}$ | 75.1$_{\pm1.2}$ | 81.6$_{\pm0.2}$ |
| ERM | ✗/✗ | 61.2$_{\pm3.6}$ | 92.0$_{\pm0.0}$ | 64.8$_{\pm1.9}$ | 82.6$_{\pm0.0}$ |

# D Experimental Details

## D.1 Complete Results

The complete results on Waterbirds, CelebA, and UrbanCars, in addition to complete results on CivilComments and MultiNLI are reported in Tables 4 and 5 respectively. The results for all methods except Group DRO + EIIL on all datasets except UrbanCars are reported by Qiu et al. [9]. The results for Group DRO + EIIL are taken from Zhang et al. [24]. Also, the results of our method and DFR are shown in Table 6

## D.2 Dominoes-Colored-MNIST-FashionMNIST

**Dominoes-Colored-MNIST-FashionMNIST (Dominoes-CMF)** is a synthetic dataset. We adopt a similar approach to previous works [37, 38, 1] using a modified version of the *Dominoes* binary classification dataset. This dataset consists of images with the top half showing CIFAR-10 images [19], divided into two meaningful classes: vehicles (airplane, car, ship, truck) and animals (cat, dog, horse, deer). The bottom half displays either MNIST [20] images from classes $\{0-3\}$ or Fashion-MNIST [21] images from classes $\{T\text{-shirt}, Dress, Coat, Shirt\}$. The complex feature (top

Table 6: A Comparison of ERM, DFR, EVaLS, and EVaLS-GL on the Dominoes-CMF Dataset. Both the worst and average of test group accuracies are presented. The mean and standard deviation are calculated based on runs with three distinct seeds.

| Method | Worst | Average |
|---|---|---|
| ERM | $50.6_{\pm 1.0}$ | $84.1_{\pm 0.0}$ |
| DFR | $60.2_{\pm 1.2}$ | $84.6_{\pm 0.4}$ |
| EVaLS-GL | $63.6_{\pm 1.3}$ | $78.7_{\pm 1.5}$ |
| EVaLS | $\mathbf{67.1_{\pm 4.2}}$ | $78.6_{\pm 2.0}$ |

half) serves as the core feature and the simple feature (bottom half) is linearly separable and correlated with the class label at 75%. Furthermore, inspired by the approaches in Zhang et al. [24], Arjovsky et al. [18], we intentionally introduce an additional spurious attribute by artificially coloring a subset of images in the following manner: 90% of the bottom half images in class $c_1$ are randomly assigned a red color, while 10% are assigned a green color, and vice versa for class $c_2$. See Table 7 for more details about the dataset statistics.

Table 7: *Dominoes-CMF* Dataset Statistics

| Top part | Bottom part | | |
|---|---|---|---|
| **CIFAR-10 Class** | **Color** | **MNIST** | **Fashion-MNIST** |
| $c_1$ (Vehicle) | Red | 13,500 | 4,500 |
| | Green | 1,500 | 500 |
| $c_2$ (Animal) | Red | 500 | 1,500 |
| | Green | 4,500 | 13,500 |
| **Total** | | 40,000 | |

Table 8: ERM Accuracies on *Dominoes-CMF* Dataset. The mean and standard deviation are reported based on three runs with different seeds.

| Top part | Bottom part | | |
|---|---|---|---|
| **CIFAR-10 Class** | **Color** | **MNIST** | **Fashion-MNIST** |
| $c_1$ (Vehicle) | Red | $99.2_{\pm 0.01}\%$ | $95.2_{1.1}\%$ |
| | Green | $84.5_{\pm 2.4}\%$ | $54.7_{\pm 0.5}\%$ |
| $c_2$ (Animal) | Red | $56.8_{\pm 5.6}\%$ | $86.7_{\pm 2.4}\%$ |
| | Green | $96.2_{\pm 0.5}\%$ | $99.3_{\pm 0.2}\%$ |

## D.3 Datasets

**Waterbirds [7]** The dataset comprises images of diverse bird species, classified into two categories: waterbirds and landbirds. Each image features a bird set against a backdrop of either water or land. Interestingly, the background scene acts as a spurious feature in this classification task. Waterbirds are primarily shown against water backgrounds, and landbirds against land backgrounds. Consequently, waterbirds on water and landbirds on land form the minority groups in the training data. It's important to note that the validation dataset for waterbirds is group-balanced, meaning birds from each class are equally represented against both water and land backgrounds. This dataset is mainly categorized as a spurious correlation dataset.

**CelebA [13]** is a widely used dataset in image classification tasks, featuring annotations for 40 binary facial attributes such as hair color, gender, and age. Hair color classification is particularly prominent in literature focusing on spurious correlation robustness. Notably, gender serves as a spurious attribute within this dataset, where a significant majority $94\%$ of individuals with blond hair

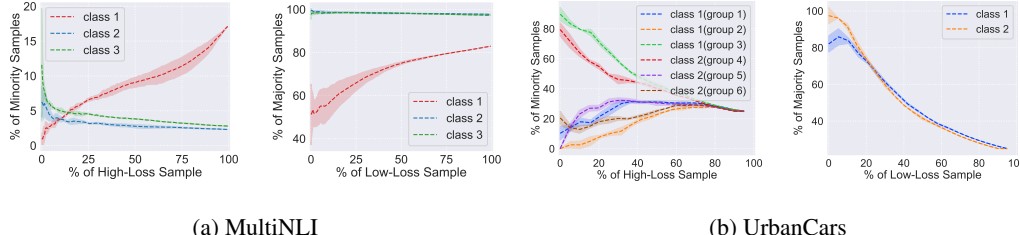

(a) MultiNLI

(b) UrbanCars

Figure 7: The proportion of minority and majority samples across different classes within various percentages of $\mathcal{D}^{LL}$ samples with highest and lowest loss for the MultiNLI (a) and UrbanCars (b) datasets. MultiNLI exhibits attribute imbalance rather than spurious correlation, which explains its different behavior compared to Waterbirds and CelebA.

are women, while men with blond hair represent a minority group. In addition to spurious correlation in the class of blond hair, this dataset also exhibits class imbalance.

**MultiNLI [15]** dataset involves a text classification task focused on determining the relationship between pairs of sentences: contradiction, entailment, or neutral. Sentences containing negation words such as "no" or "never" are under-represented in all three classes, inducing attribute imbalance in the dataset. Figure 7 illustrates the distinct behavior of this dataset compared to other datasets that contain spurious attributes.

**CivilComments [16]** dataset, as part of the WILDS benchmark, involves a text classification task focused on labeling online comments as either "toxic" or "not toxic". Each comment is associated with 8 attributes, including gender (male, female), sexual orientation (LGBTQ), race (black, white), and religion (Christian, Muslim, or other), based on whether these characteristics are mentioned in the comment. While there is a small attribute imbalance in the dataset, it can categorized into datasets with class imbalance. In this paper, we use the implementation of the dataset by the WILDS package [39].

**UrbanCars [14]** is an image classification dataset with multiple shortcuts. Each image in the dataset consists of a car in the center of the image on a natural scene background, with another object to the right of the image. Images are labeled *Urban* or *City* according to the type of car present in the center. However, each of the backgrounds and the additional objects is highly correlated with the label. While the test set consists of 8 environments based on combinations of the core and two spurious patterns, the training and validation set consist of four groups, based on combinations of the label and only one of the shortcuts.

### D.4 Training Details

**ERM** For Waterbirds and CelebA, we utilize the ResNet50 checkpoints available in the GitHub repository of Kirichenko et al. [1] as our base model. We use the ResNet-50 architecture provided by the `torchvision` package. In the case of Civil-Comments and MultiNLI, we adopt a similar approach to Kirichenko et al. [1], using `BertForSequenceClassification.from_pretrained('bert-base-uncased', ...)` from the `transformers` package. The model is trained using the AdamW optimizer with a learning rate of $10^{-5}$, weight decay of $10^{-4}$, and a batch size of 16 for a total of 5 epochs.

For the UrbanCars dataset, we adhere to the settings described in Li et al. [14], which involves training a ResNet-50 model pretrained on ImageNet using the SGD optimizer with a learning rate of $10^{-3}$, momentum of 0.9, weight decay of $10^{-4}$, and a batch size of 128 for 300 epochs. For the Dominoes-CMF dataset, we train a ResNet18 model pretrained on ImageNet for 20 epochs with a batch size of 128 and an SGD optimizer with a learning rate of $10^{-3}$, momentum of 0.9, and weight decay of $10^{-4}$.

**EVaLS and EVaLS-GL** For every dataset, EIIL was utilized with a learning rate of $0.01$, a total of 20000 steps, and a batch size of 128. The last layer of the model was trained on all datasets using the

Adam optimizer. A batch size of 32 and a weight decay of $10^{-4}$ were used for all datasets. Our method was evaluated on the validation sets of each dataset, considering both fine-tuning and retraining of the last layer. For all datasets, with the exception of MultiNLI, retraining provided superior validation results. The specifics regarding the number of epochs and the ranges for hyperparameter search (including learning rate, $l_1$-regularization coefficient ($\lambda$), and the number of selected samples ($k$)) for each dataset are as follows:

- **Waterbirds**.
    - epochs = 100,
    - lr = $5 \times 10^{-4}$,
    - $\lambda \in \{0, 0.01, 0.02, 0.03, 0.04, 0.05, 0.06, 0.07, 0.08, 0.09, 0.1, 0.2, 0.3, 0.4, 0.5\}$,
    - $k \in \{20, 25, 30, 35, 40, 45, 50, 55, 60\}$.

- **CelebA**
    - epochs = 50,
    - lr = $5 \times 10^{-4}$,
    - $\lambda \in \{0, 0.01, 0.02, 0.03, 0.04, 0.05, 0.06, 0.07, 0.08, 0.09, 0.1, 0.2, 0.3, 0.4, 0.5,$
      $0.6, 0.7, 0.8, 0.9, 1, 2\}$,
    - $k \in \{50, 100, 150, 200, 250, 300\}$.

- **UrbanCars**
    - epochs = 100,
    - lr $\in \{5 \times 10^{-4}, 10^{-3}\}$,
    - $\lambda \in \{0, 0.01, 0.02, 0.05, 0.1, 1\}$,
    - $k \in \{10, 20, 30, 50, 63\}$.

- **CivilComments**
    - epochs = 50,
    - lr = $5 \times 10^{-4}$,
    - $\lambda \in \{0, 0.01, 0.02, 0.03, 0.04, 0.05, 0.06, 0.07, 0.08, 0.09, 0.1, 0.2, 0.3, 0.4, 0.5,$
      $0.6, 0.7, 0.8, 0.9, 1, 2\}$,
    - $k \in \{500, 750, 1000, 1250, 1500\}$.

- **MultiNLI**
    - epochs = 200,
    - lr $\in \{10^{-2}, 10^{-3}\}$,
    - $\lambda \in \{0, 0.01, 0.02, 0.03, 0.04, 0.05, 0.06, 0.07, 0.08, 0.09, 0.1, 0.2, 0.3, 0.4, 0.5\}$,
    - $k \in \{20, 30, 40, 50, 60, 75, 100, 125, 150, 200, 250, 300\}$.

# E  Ablation Study

## E.1  Use of EIIL with DFR and AFR

We conducted an ablation study to investigate the impact of using environments inferred from EIIL on model selection. Specifically, we benchmarked the performance of DFR and AFR with EIIL-inferred groups. The results, presented in Table 9, demonstrate the effectiveness of incorporating EIIL-inferred groups in model selection. The results show that while EIIL-inferred groups reduce the performance compared to ground-truth annotations for model selection, they still can be effective for robustness to an extent. Moreover, EVaLS outperforms these two methodw when using EIIL inferred environments.

Table 9: Results of DFR and AFR with EIIL-inferred environment for model selection.

| Method | Waterbirds | Celeba |
|---|---|---|
| DFR (with EIIL) | $\mathbf{92.21 \pm 0.02}$ | $\mathbf{85.55 \pm 1.0}$ |
| AFR (with EIIL) | $82.6 \pm 0.04$ | $72.5 \pm 0.01$ |

## E.2  Other Group Inference Methods

In addition to EIIL, other group inference methods could be utilized for partitioning the model selection set into environments.

**Error Splitting**  JTT [5] partitions data into two correctly classified and misclassified sets based on the predictions of a model trained with ERM. We split each of these two sets based on labels of samples, obtaining $|\mathcal{Y}| \times 2$ environments.

**Random Classifier Splitting**  uses a random classifier to classify features obtained from a model trained with ERM into correctly classified and misclassified sets. Similar to error splitting, we split the sets based on group labels. The difference between error splitting and random classifier splitting is solely in the reinitialization of the classification layer.

The results for EVaLS-ES (EVaLS+Error Sampling) and EVaLS-RC (EVaLS+Random Classifier) are shown in Table 10. One limitation of error splitting is that in datasets with noisy labels or corrupted images, samples that an ERM model misclassifies may not always belong to minority groups. In these situations, choosing models based on their accuracy on corrupted data could lead to the selection of models that are not robust to spurious correlations. This is demonstrated by the results of EVaLS-ES on the CelebA dataset.

This shortcoming of error splitting can be alleviated by employing a random classifier instead of the ERM-trained one. Due to the feature-level similarity between minority and majority samples in datasets affected by spurious correlation [23, 1, 29], it is expected that the classifier can differentiate between the groups to some extent. As shown in Table 10, surprisingly, EVaLS-RC produces results that are generally comparable to EVaLS. However, the performance of this method may have high variance, depending on the different initializations of the classifier.

Table 10: The performances of three environment inference methods, when combined with loss-based sample selection, are evaluated on spurious correlation benchmarks. The mean and standard deviation values are calculated over three separate runs, each initiated with a different seed.

| Method | Waterbirds | | CelebA | | UrbanCars | |
|---|---|---|---|---|---|---|
| | Worst | Average | Worst | Average | Worst | Average |
| EVaLS-ES | $82.1_{\pm 1.2}$ | $\mathbf{94.3_{\pm 0.04}}$ | $48.4_{\pm 11.6}$ | $69.5_{\pm 6.5}$ | $79.2_{\pm 2.9}$ | $86.1_{\pm 0.9}$ |
| EVaLS-RC | $\mathbf{88.7_{\pm 1.0}}$ | $94.3_{\pm 1.1}$ | $78.1_{\pm 5.1}$ | $\mathbf{93.5_{\pm 0.2}}$ | $\mathbf{82.4_{\pm 3.2}}$ | $\mathbf{88.2_{\pm 0.8}}$ |
| EVaLS | $88.4_{\pm 3.1}$ | $94.1_{\pm 0.1}$ | $\mathbf{85.3_{\pm 0.4}}$ | $89.4_{\pm 0.5}$ | $79.4_{\pm 3.1}$ | $86.5_{\pm 1.5}$ |

## F Societal Impacts

Real-world datasets often encapsulate social biases that stem from entrenched stereotypes and historical discrimination, affecting various groups such as genders and races. Machine learning methods, which learn the correlation between patterns in input data and their targets (e.g., labels in a classification task) [40], inadvertently absorb this bias. This unintended consequence leads to fairness issues in many applications. While strategies to mitigate such biases have been proposed (as discussed comprehensively in Section A), societal biases are not always known and determined. We believe that our work, as it addresses these unidentified biases, takes a significant step towards making machine learning fairer for our society.

## G Computational Resources

Each experiment was conducted on one of the following GPUs: NVIDIA A100 with 80G memory, NVIDIA Titan RTX with 24G memory, Nvidia GeForce RTX 3090 with 24G memory, and NVIDIA GeForce RTX 3080 Ti with 12G memory.

