# OpenReview forum: "Trained Models Tell Us How to Make Them Robust to Spurious Correlation without Group Annotation"
_NeurIPS.cc/2024/Conference — Submitted to NeurIPS 2024_

### Official Review · Reviewer_7E4z · 2024-07-01

**Soundness:** 2
**Presentation:** 3
**Contribution:** 2
**Rating:** 5
**Confidence:** 4

**Summary:**

This paper addresses the problem of subpopulation generalization, also known as spurious correlations. Building on the Last Layer Retraining (DFR) method, it removes the constraints on a small subset of annotations. The paper introduces the Environment-based Validation and Loss-based Sampling (EVaLS) method. Unlike DFR, EVaLS divides the validation set $D^{val}$ into two parts: (1) $D^{LL}$,  where losses from an ERM-trained model are used as a proxy for identifying minority groups for retraining, and (2) $D^{MS}$, where environment inference methods are used for partitioning environments. The paper presents theoretical insights and empirical results demonstrating the effectiveness of EVaLS.

**Strengths:**

* The paper is well-structured and presented in a clear and organized manner, making it easy to comprehend and follow along.
* The proposed method is simple but effective and explores a relatively challenging area in existing literature (*i.e.* subgroup generalization without group annotations).
* The authors provide some theoretical analysis to support their claims.

**Weaknesses:**

* The novelty and contribution of the proposed method may be limited for the following reasons: 1) The paper combines multiple previously proposed methods (*i.e.* DFR [1], EIIL [2]) all at once, which inherently guarantees a nontrivial performance; (2) The primary technical contribution, at least from my perspective, is the loss-based sampling, which has been already explored extensively in the noisy label literature and has been used as tools for pseudo-labeling.
* The paper fails to discuss recently proposed methods that also require no group annotations, such as SELF [3], BAM [4], and BPA [5]. In particular, SELF is also a direct follow-up of DFR. The authors are encouraged to discuss the limitations and strengths of loss-based schemes against the class-based schemes advocated by SELF and BAM.
* More analyses can be included to provide further understanding of the selected loss-based samples. For example, given a threshold, how much percent of the high-loss and low-loss samples are indeed the minority and majority samples and how does this percentage change with the threshold?

[1] Last Layer Re-Training is Sufficient for Robustness to Spurious Correlations, ICLR 2023

[2] Environment inference for invariant learning. ICML 2021

[3] Towards Last-layer Retraining for Group Robustness with Fewer Annotations. NeurIPS 2023

[4] Bias Amplification Enhances Minority Performance. TMLR 2024

[5] Unsupervised learning of debiased representations with pseudo-attributes. CVPR 2022

**Questions:**

* Is EVaLS sensitive to hyperparameters?
* [minor] There seem to be typos in your corollary C.1.
* Can the authors make clarifications on how the conditions in proposition 3.1 are met or relaxed in practice? How does the distribution of actual experimental benchmark datasets compare to your assumptions?

**Limitations:**

Aforementioned in Weaknesses and Questions.

---

> ### Author Rebuttal · Authors · 2024-08-07
>
> Thank you for highlighting the relevant works and for your insightful requests regarding the method’s sensitivity and the compatibility of its theoretical and practical aspects.
> # Weakness 1
> Please refer to the general response in the Author Rebuttal for a review of the contributions of this work. We also note that the modules in the method can be replaced while completely maintaining the overall contribution and framework. In Appendix E, we substitute loss-based sampling with other methods (Sec. E.1) and provide an ablation study on replacing EIIL with other simple environment inference techniques (Sec. E.2).
> # Weakness 2
> Thank you for pointing out relevant works.
>
> As stated in Appendix A (L552-555), SELF introduces CB last-layer retraining that does not require group annotations for retraining or model selection. Another method proposed by SELF, ES disagreement, requires group annotations for model selection but not for the retraining (see Appendix A, L547-549). It also needs an early-stopped version of the trained ERM along with the final model.
>
> BAM is a two-stage training method that works with or without group annotations for model selection. Without group annotations, BAM uses ClassDiff to select hyperparameters, ensuring balanced performance across all classes.
>
> BPA is a clustering-based method (see L206). While BAM claims BPA doesn’t need group annotations for model selection, it is important to note that our investigation shows that it does.
>
> All comparisons use similar splits and architectures as the benchmarked methods in the paper.
>
> ## Coamparison with class-balanced schemes
> Following Yang et al. (reference [4] in the paper): Given input $x=(x_{c}, x_{s})\in\mathcal{X}$, which consists of core feature $x_{c}$ and spurious feature $x_{s}$, and label $y\in\mathcal{Y}$, we can write the classification model:$$P(y|x)=\frac{P(x|y)}{P(x)}P(y)=\frac{P(x_{c},x_{s}|y)}{P(x_{c},x_{s})}P(y)=\frac{P(x_{c}|y)}{P(x_{c})}\frac{P(x_{s}|x_{c}, y)}{P(x_{s}|x_{c})}P(y)$$Spurious correlation occurs when $P(x_{s}|x_{c}, y)\gg P(x_{s}|x_{c})$, while class imbalance represents the scenario where $P(y)\gg P(y')$ for $y, y'\in\mathcal{Y}$. Thus, as stated in Preliminaries section (L112-123), such shifts can occur independently in datasets.
>
> Yang et al. observes in Figure 2 of the paper that Waterbirds, CelebA, and CivilComments have significant class imbalance, but MultiNLI does not. It can be seen that robustness improvements in class-balanced schemes in SELF and BAM relate to the amount of class imbalance in the dataset. For example, as emphasized in various parts of the paper (L79, L257, L313-315), the CivilComments dataset does not show spurious correlation but does exhibit class imbalance. Consequently, class-balanced schemes significantly improve worst-group accuracy on this dataset. However, results for MultiNLI (which exhibits attribute imbalance but no class imbalance (see L114-116)) are just slightly higher than ERM (for BAM) or even lower than it (for SELF).
>
> Class-balanced schemes are helpful for handling class-imbalance-based subpopulation shifts but fail in addressing spurious correlations. As the authors of SELF note, these schemes can’t match state-of-the-art methods. This is due to their inability to manage subpopulation shifts caused by spurious correlations. Our approach, which involves selecting an equal number of samples from each group through a loss-based sampling scheme, creates a class-balanced retraining dataset. However, our paper clarifies (L316-325) that environment-based validation doesn’t address other types of subpopulation shifts beyond spurious correlations.
>
> Worst group and average accuracy (in parentheses) for CB last-layer retraining (SELF) and BAM on datasets with spurious correlations is as follows.
>
> |Method|Waterbirds|CelebA|UrbanCars|
> |-|-|-|-|
> |CB last-layer retraining|$92.6_{\pm0.8}(94.8_{\pm0.3})$|$73.7_{\pm2.8}(93.6_{\pm0.2})$|$21.9_{\pm13.0}(49.1_{\pm7.0})$|
> |BAM + ClassDiff|$89.1_{\pm0.15}(91.4_{\pm0.31})$|$80.1_{\pm3.32}(88.4_{\pm2.32})$|-|
>
> Although we couldn’t provide BAM results for UrbanCars due to high training computational costs, it’s evident that on this benchmark, which lacks class imbalance but has spurious correlations, the reported class-balanced scheme shows no improvement compared to ERM.
> ## Discussing the results of the suggested methods
> Regarding UrbanCars: we were unable to provide results for BAM during the rebuttal phase due to its significant computational cost. For ES disagreemnet SELF, WGA and AA are $82.1_{\pm1.8}$ and $89.3_{\pm0.8}$ respecitvely. For BPA, WGA is $62.4$.
>
> For other datasets, results are obtained from the respective papers.
>
> For Waterbirds, ES disagreement SELF and BAM have higher WGA at the cost of lower average accuracy. EVaLS-GL outperform these methods with similar level of group supervision on all other datasets.
> # Weakness 3
> Figure 2 in the paper provides exactly the information you’re looking for! Please refer to the general response.
> # Q1
> Please see the general response.
> # Q2
> You are absolutely correct. The $F^2(\beta)$ should change to $F^2(\alpha)$ as $tail_L$ is a function of $\alpha$.
> # Q3
> Due to the technical difficulty of dealing with arbitrary distributions, our proof relies on the Gaussian nature of the logit distribution. In a case study (class 2 of CelebA with 94% spurious ratio on 20000 datapoints), this assumption almost holds (see qqplot in Figure 4 in the attached PDF). Evaluating our theorem's conditions, the empirical variance of the minority is higher than the majority (see “Distribution of logits” in the general rebuttal), satisfying the necessary and sufficient condition (L619) for Proposition C.1. The theorem proves the existence of suitable $\alpha, \beta$ (see L223-225). To empirically justify our tail-based method, we tested all possible tails $\alpha, \beta$ and found 6 value pairs that make the dataset group balance. Thus, our practical results align with our theoretical work.

---

> > ### Comment · Reviewer_7E4z · 2024-08-07
> > **Thanks for your response**
> >
> > Thank you for your detailed response. I think it generally addresses my concerns so I will increase my score. I would encourage the authors to include these additional discussions in the final version of the paper.

---

> ### Author Response · Authors · 2024-08-07
> **Response to Reviewer 7E4z**
>
> We deeply appreciate your consideration of our rebuttal. Thank you for suggesting the inclusion of the discussions in the final version of the paper; we will certainly do so.
>
> We have carefully addressed all your concerns based on your feedback. If there are any specific areas where you feel further clarification or justification is needed, we are eager to address them promptly. Your guidance helps us achieve the highest standards, and we hope this will be reflected in a higher score.

---

### Official Review · Reviewer_SN9B · 2024-07-11

**Soundness:** 2
**Presentation:** 3
**Contribution:** 2
**Rating:** 6
**Confidence:** 3

**Summary:**

To address the issue of spurious correlations when group labels are unavailable, this paper proposes a new method called EVaLS. It first creates a balanced training dataset using loss-based sampling. Then, it evaluates the accuracy of the balanced training set based on the inferred environments from the validation set, and selects models accordingly.

**Strengths:**

1. The paper is well-written, and includes a rich set of experiments with necessary theoretical explanations.

2. It is essential to discuss the multiple  (unknown) spurious features case which has been overlooked in previous studies.

**Weaknesses:**

1. Why is the approach of using high-loss points (considered as the minority group) more effective than directly using misclassified points (considered as the minority group) in methods like JTT? Intuitively, compared to misclassified points, high-loss points are more "implicit" and no obvious thresholds, which could potentially result in high-loss points actually belonging to the majority group, thus exacerbating the imbalance in resampling.

2. If the author can show the balance level of samples obtained through loss-based sampling compared to directly using labels (misclassified points), it could further illustrate the advantages of loss-based sampling.

3. In Section "Mitigating Multiple Shortcut Attributes", if color is treated as a known spurious attribute and shape as an unknown spurious attribute, how would the performance of EVaLS be affected? Based on my understanding, there is a possibility that simplicity bias could cause the model to prioritize learning the simpler feature, color, and struggle to learn the more complex shape attribute. Therefore, considering color as known and shape as unknown can better show the performance of EVaLS in handling complex spurious features.

**Questions:**

See weaknesses.

**Limitations:**

See weaknesses.

---

> ### Author Rebuttal · Authors · 2024-08-07
>
> Thank you for your constructive and insightful feedback. Our response to the mentioned weaknesses is as follows:
>
> # Weakness 1
>
> 1. We must emphasize that the optimal number of selected high/low loss samples for retraining the last layer is chosen from a set of various values, based on the worst validation group/environment accuracy. Hence, the proposed method could be considered a more general solution that encompasses selecting misclassified samples (If selecting misclassified samples yields the best validation results, this configuration is chosen automatically by our method).
> 2. Having a hyperparameter ($k$) for controlling the number of selected samples from loss tails gives us flexibility to handle some challenges. For example,
> - There is a tradeoff between the purity and the number of selected high-loss samples: as observed in Figures 2 and 7 in the paper, minority samples are more commonly seen among samples on which the ERM model has a high loss. As we increase the number of high-loss selected samples, the proportion of minority samples among them decreases. On the other hand, retraining on a larger number of samples could improve the overall training process. Choosing flexibly among various numbers of selected samples grants EVaLS the advantage of finding an optimal point in this tradeoff. You can see how sensitive the worst validation group accuracy is to selecting $k$ in Figure 5 in the attached PDF in the general response. Choosing only misclassified samples could not handle this trade-off especially when the number of misclassified samples is too high or low.
> - There are situations in which the dataset contains corrupted data or samples with label noise on which the ERM model has a high loss or may misclassify. Using such samples for retraining the last layer degrades its performance. However, choosing the number of selected samples based on the worst validation group/environment accuracy instead of using a specific number allows our method to mitigate this issue to an extent.
>
> We also conducted an experiment in which we replaced the loss-based sampling in EVaLS with the selection of misclassified samples and an equal number of randomly selected correctly classified samples from each class. As can be seen in the results, the performance is degraded compared to EVaLS on the Waterbirds and Urbancars datasets, with a slight improvement but a higher standard deviation on the CelebA dataset.
>
> | | Waterbirds| CelebA| Urbancars|
> |:------:|:-------:|:--------:|:--------:|
> | | Worst/Average| Worst/Average| Worst/Average|
> |Misclassified Selection | $77.8_{\pm 5.2}$/$94.0_{\pm 0.4}$   | $85.9_{\pm 1.0}$/$89.4_{\pm 0.8}$ | $78.4_{\pm 4.5}$/$86.9_{\pm 1.4}$    |
> | EVaLS| $88.4_{\pm 3.1}$ /$94.1_{\pm 0.1}$ | $85.3_{\pm 0.4}$/$89.4_{\pm 0.5}$ | $82.13_{\pm 0.92}$\|$88.1_{\pm 0.9}$ |
>
>
> # Weakness 2
> As requested, the balance levels of samples selected based on loss and misclassification are presented in the following table for the Waterbirds, CelebA, and UrbanCars datasets.
> The ratio $\frac{minority}{k}$​ for each class is reported for three seeds, with the average across all seeds reported in parentheses. The results for the UrbanCars dataset are reported for the minority group with the lowest number of training samples.
>
> |Dataset|High-Loss| |Misclassified | |
> |-|:---:|:----:|:----:|:---:|
> | | Class 1 | Class 2 | Class 1 | Class 2 |
> | Waterbirds | $\frac{53}{55} , \frac{44}{45}, \frac{25}{25}(98_{\pm 1.0})$  | $\frac{48}{55}, \frac{41}{45}, \frac{21}{25}(87.5_{\pm 3.6})$|$\frac{40}{41},\frac{40}{41}, \frac{40}{41} (97.6_{\pm 0.0})$|$\frac{17}{20}, \frac{17}{20}, \frac{17}{20} (85.0_{\pm 0.0}) $ |
> |CelebA| - | $\frac{58}{250}, \frac{58}{250}, \frac{20}{50}(28.8_{\pm 9.7})$ | - | $\frac{58}{262},\frac{58}{250},\frac{58}{270} (22.3_{\pm 0.8})$ |
> |Urbancars| $\frac{9}{10}, \frac{24}{30}, \frac{24}{30} (83.3_{\pm 6.0})$ | $\frac{9}{10}, \frac{20}{30}, \frac{19}{30} (73.3_{\pm 14.5})$  | $\frac{44}{66},\frac{49}{73}, \frac{48}{69} (67.8_{\pm 1.6})$ | $\frac{35}{64}, \frac{29}{50}, \frac{33}{55} (57.6_{\pm 2.7})$|
>
>
> # Weakness 3
>
> Thank you for requesting such an interesting comparison! We believe that your intuition is correct. When the unknown spurious attribute is shape instead of color, methods that use the known group annotations show higher worst group accuracy compared to the previous scenario. In other words, when the unknown spurious attribute is a weaker shortcut, group robustness becomes a simpler task. Note that since EVaLS does not rely on any information regarding spurious attributes, it does not matter which spurious attribute is known and which is unknown; the results remain the same as reported in Table 2 of the paper. You can see the results for other methods for the new settings in the table below.
>
> |Method|Worst Group Accuracy|
> |-|:-:|
> |AFR|$60.97_{\pm2.64}$|
> |AFR + EIIL|$61.54_{\pm1.85}$|
> |DFR|$71.5_{\pm3.2}$|
> |EVaLS-GL|$65.5_{\pm0.6}$|
>
> We found that utilizing group annotations for a simpler attribute (a stronger shortcut) enhances results compared to the previous setting where the unknown attribute was simpler. Regarding DFR, we believe that part of its performance is due to the larger number of retraining samples compared to EVaLS and EVaLS-GL (as stated in l355), rather than the provided group annotation. To verify this, we evaluated the performance of DFR when its number of retraining samples is reduced to the number of samples used for EVaLS. In this case, the performance of DFR drops to $66.3_{\pm1.15}$, which is lower than EVaLS.
>
> Additionally, the performance gap between AFR and AFR + EIIL, as well as between EVaLS-GL and EVaLS (using environment-based validation instead of ground truth validation group labels of the known spurious attribute), decreases.

---

> > ### Comment · Reviewer_SN9B · 2024-08-10
> >
> > Dear author,
> >
> > Thank you for your reply and the additional experimental results.
> >
> > After reading the reply and the opinions of other reviewers, I maintain my score. Although I agree with Reviewer 7E4z's point that this work is of limited novelty, I think the author's research is complete and detailed with enough experiments to show the effectiveness of EVaLS. In addition, I appreciate the discussion on multiple (unknown) spurious features. Therefore, I keep my score.
> >
> > Best,
> >
> > Reviewer SN9B

---

> > > ### Author Response · Authors · 2024-08-11
> > >
> > > Dear Reviewer SN9B,
> > >
> > > Thank you for taking the time to read our rebuttal. We are pleased to hear that you find our research complete and detailed, with sufficient experiments demonstrating its effectiveness.
> > >
> > > We must emphasize that the title of our work reflects its content. We propose a method to derive all necessary information from the model itself to enhance its robustness against spurious correlations. Section 3, which introduces EVaLS, presents solutions such as loss-based sampling for creating a balanced dataset and partitioning the validation set into environments for model selection. We have also conducted multiple ablations in Section E to evaluate different modules while maintaining the overall framework. Thus, our work has never proposed using and combining previous methods as a novelty (as Reviewer 7E4z has stated).
> > >
> > > Our work demonstrates the following points, as stated at the beginning of the general author rebuttal:
> > >
> > > - In contrast to what is considered a requirement for previous methods (see L528-549 in Appendix A-Related Work), we show that ideal group discovery is not required for model selection in robustness to spurious correlations. Instead, identifying environments with group shifts (Sec. 3.2) works effectively (see Table 1).
> > > - Relying on what a trained model learns, rather than auxiliary (even ground-truth) information (as in methods like AFR, DFR, etc.), could be more effective in the case of unknown spurious correlations (see Table 2 and Table 1-UrbanCars).
> > > - Loss-based sampling (Sec. 3.1) is not only an effective method for robustness to group shifts (compared to other methods such as loss-based weighting schemes like AFR or upweighting misclassified samples like JTT; see EVaLS and EVaLS-GL in Table 1), but it is also supported by a theory for data balancing with general assumptions (Sec. 3.3 - Theoretical Analysis).
> > >
> > > Best regards,
> > > Authors of Submission 19478

---

### Official Review · Reviewer_RE7s · 2024-07-12

**Soundness:** 3
**Presentation:** 2
**Contribution:** 2
**Rating:** 6
**Confidence:** 4

**Summary:**

The paper studies how to improve the model’s robustness to multiple spurious correlations when the group labels (indicator for spurious correlation) are unknown in general. The proposed approach, EVaLS, leverages the loss from a base ERM model to sample a balanced subset to prevent learning from spurious correlations. In addition, a new synthetic dataset (Dominoes-CMF) for multiple spurious attributes is crafted. Empirically, the proposed approach sometimes has advantages over the rest of the baselines when using the same amount of additional information (group label).

**Strengths:**

1. The main paper is generally well-written.
2. The theoretical analysis in Section 3.3 (with derivations and proofs in Appendix) shows that for one-dimensional Gaussian distributions, choosing the tails on the two sides of the distributions creates balanced groups, even though the original data distribution is skewed.
3. Environment inference technique is demonstrated to be useful for separating the dataset into groups with different distributions of the subpopulations and then for model selection.
4. The proposed technique only requires last-layer retraining on part of the validation set, which is generally more efficient.

**Weaknesses:**

1. Figure 2 attempts to illustrate more minority samples have high loss while the majority samples have low loss. However, in each of the plots, only the % of one of the minority or majority groups is shown. The illustration can be improved by showing the % of both majority and minority groups in the same plot, and showing the actual distribution of the loss for the groups.
2. Though the idea is straightforward, it is unclear how the loss-based instance sampling is actually implemented. It is helpful to provide an algorithm or pseudocode to improve the presentation.
3. The theoretical analysis is generally sound but limited to a case without discussing the use of the loss (which may not be Gaussian) and the spurious correlations (which involve at least two dimensions of core and spurious features [1]).
4. The experimental results are less polished and sometimes the advantages are not so clear over other baselines. Some results are missing for datasets such as UrbanCars and MultiNLI. Only a few baselines are compared for the new dataset in Table 2. There is also no convincing and fine-grained analysis (e.g., ablation study) to understand how the proposed approach ensures data balancing and improves group robustness.
5. The paper initially focuses on improving group robustness when multiple spurious correlations are present, but the experimental results are lacking for these more challenging datasets.

[1] Sagawa, Shiori, Aditi Raghunathan, Pang Wei Koh, and Percy Liang. "An investigation of why overparameterization exacerbates spurious correlations." In *International Conference on Machine Learning*, pp. 8346-8356. PMLR, 2020.

**Questions:**

1. Since it’s an essential prerequisite for loss-based sampling, how well are the majority/minority groups separated in the logit space?
2. How is the number of balancing samples $k$ chosen? How balanced are the samples when they are selected from the optimal $k$?
3. I am curious about why the result for Civilcomment dataset is so much higher than the other baselines. Is the evaluation consistent with the baseline methods?
4. Minor typo: line 233 should be without loss of generality (w.l.o.g.).
5. Other minor issues with the experiments: The best-performing results in each category of approaches should be bold. The column header “best” should be “average” in Tables 4 and 5.

**Limitations:**

The authors have discussed the limitations in Section 5.

---

> ### Author Rebuttal · Authors · 2024-08-07
>
> Thank you for your detailed reviews and insightful questions.
> # Weakness 1
> Figure 2 in the paper **does not** show the proportion of minority (majority) samples that have high (low) loss. Instead, it depicts the proportion of minority/majority samples among the top x% of samples with the highest/lowest loss values. At the 100% mark on the x-axis, it shows the proportion of minority/majority groups in the entire dataset.
>
> For example:
> - In the second plot from the left, about 80% of the 50% samples with lowest loss in both class 1 and 2 of the Waterbirds dataset belong to majority groups.
> - In the second plot from the right, over 30% of the highest-loss samples (top 1%) in the unbalanced class (class 2) of CelebA belong to minority groups.
>
> When n% of samples in the top x% of samples with the highest/lowest loss belong to minority/majority groups, the remaining (100-n)% belong to majority/minority groups. To clarify, we will update the x-axis title to “% of selected samples from samples with highest/lowest loss values”.
>
> As requested, Figure 1  in the attached PDF of the general author’s rebuttal shows the % of both majority and minority groups in the same plot, and Figure 3 illustrates the loss value distribution for these groups.
>
> Please let us know if any further information or clarification is required.
> # Weakness 2
> We appreciate the suggestion to include pseudocode. We present a pseudocode in the “General Response”.
> # Weakness 3
> The scenario you described, involving multiple dimensions for spurious and core features, already aligns perfectly with our framework. As noted in L217-219 (and elaborated further in L585-589 in the Appendix C), we assume that both minority and majority groups follow a Gaussian distribution in the *feature space*. This assumption leads to Lemma C.1 (line 590), where the Gaussian distribution for our logits (**not losses**) is derived. Using a Gaussian distribution in the feature space is a common practice due to the technical challenges associated with other distributions. The paper you referenced also assumes a Gaussian distribution for both core and spurious dimensions. So, we believe that our assumption contains the one you have mentioned.
>
> As mentioned in L215-216, the order of samples in logit space and loss space is monotonic. Therefore, it is unnecessary to focus on the distribution of the loss or assume any specific distribution (Gaussian or anything else) for it.
> # Weakness 4
> **”results are missing for datasets such as UrbanCars and MultiNLI.”**
>
> The missing results (depicted by - in Table 1 in the paper) are as follows:
>
> |Method|Dataset|Worst Group Accuracy|Average Accuracy|
> |-|:-:|:-:|:-:|
> |GDRO|UrbanCars|$73.1_{\pm2.0}$|$84.2_{\pm1.3}$|
> |GDRO + EIIL|UrbanCars|$76.5_{\pm2.6}$|$85.4_{\pm2.1}$|
> |GDRO + EIIL|MultiNLI|$61.2_{\pm0.5}$|$79.4_{\pm0.2}$|
> |JTT|UrbanCars|$79.5_{\pm5.1}$|$86.3_{\pm1.0}$|
>
> **“Only a few baselines are compared for the new dataset in Table 2.”**
>
> Table 2 shows that loss-based sampling and environment-based validation effectively handle unknown spurious correlations, focusing on methods with similar setups (last-layer retraining). We also include the worst group accuracy for a new method (AFR) and its environment-based validation version. The worst group accuracies are:
>
> AFR: $54.27_{\pm2.73}$
>
> AFR + EIIL: $61.54_{\pm1.85}$
>
> **“There is also no convincing and fine-grained analysis (e.g., ablation study) to understand how the proposed approach ensures data balancing and improves group robustness.”**
>
> *Regarding data balancing*:
> Figure 2 in the paper shows the proportion of minority/majority samples among those with the highest/lowest loss at various thresholds. The balance level for chosen \( k \) is detailed in the response to Q2 in this rebuttal.
>
> *Regarding group robustness*:
> We conducted several ablation studies and analyses to understand how EVaLS work:
> - Group shifts under environment inference (Appendix-Table 3)
> - Ablations on environment-based validation methods (Sec. E.2) and alternatives to loss-based sampling (Sec. E.1)
> - Experiments on benchmarks with single and multiple spurious correlations (e.g., UrbanCars)
> - A new dataset (Dominoes-CMF) for studying multiple independent spurious correlations
>
> For further analysis, please let us know.
>
> # Weakness 5
> Regarding multiple spurious correlation dataset, we use UrbanCars (as outlined in L308-312), and also propose Dominoes-CMF dataset with two independent spurious correlations (Sec. 2.2). Table 2 shows that EVaLS, which doesn’t require group annotation, improves model robustness to unknown spurious correlations compared to more supervised solutions.
>
> Finally, note that EVaLS is a robustification method to unknown spurious correlations that are learnt by a trained model. So, as illustrated in the results of the paper, it improves robustness for various benchmarks with single or multiple spurious correlations.
> # Q1
> Please refer to the “Distribution of logits for training datasets” section in the general response.
> # Q2
> The optimal number of balancing samples, $k$, is determined by the worst accuracy on inferred environments. The tables below report the proportion of minority group in selected high-loss samples, as well as the proportion of majority group within the selected low-loss samples for a sample seed:
>
> Waterbirds
> |Class 1 (#Min/#High loss)|Class 1 (#Maj/#Low loss)|Class 2 (#Min/#High loss)|Class 2 (#Maj/#Low loss)|
> |:-:|:-:|:-:|:-:|
> |$44/45$|$38/45$|$41/45$| $42/45$|
>
> Class 2 of CelebA
> |#Min/#High loss|#Maj/#Low loss|
> |:-:|:-:|
> |$58/250$| $249/250$|
>
>
> # Q3
> Refer to L313-315. The CivilComments dataset has class imbalance, so fine-tuning/retraining on class-balanced data can significantly improve group robustness. EVaLS-GL’s effectiveness is due to using a class-balanced dataset for retraining, as also noted in reference [2] (see L553-554).
> # Q4 and Q5
> Thanks for informing us about the typos. You are right. We will correct them in the final version.

---

> > ### Comment · Reviewer_RE7s · 2024-08-09
> > **Reviewer Response**
> >
> > Thanks for providing the detailed responses including experimental results, pseudo code, useful statistics, etc.
> >
> > W3: To clarify, what I meant was that the proposed approach utilizes “**loss**-based sampling” (class-dependent), but the analysis in proposition 3.1 is based on the **logits** (class-independent). It’s perfectly fine to assume Gaussian distribution in the features space and hence logit space, but when the loss is involved, it becomes a bit more complicated. This is because now we have different classes even for logistic regression. The logistic loss is monotonic for each class but when the label flips it’s no longer the same function. As assumed there are two Gaussian distributions for the **logits**, for binary classification, there will be four resulting distributions on the **losses**, because there are two transformations $-log(\sigma(L))$ and $-log(1-\sigma(L))$. Does the analysis for logits still hold in this scenario with losses?
> >
> > Furthermore, I referenced [1] because they considered two feature dimensions (core and spurious), the majority/minority groups are clearly defined as spuriously-correlated/non-spuriously-correlated w.r.t. the features and labels. When you reduce the analytical setup to majority and minority distributions, the relationship with spurious correlation becomes blurred. That’s why I mentioned it as a limitation.
> >
> > W4: Thanks for filling in the missing results. Please also include the few baselines pointed out by reviewer 7E4z to make the empirical comparison more comprehensive, but just to clarify why CivilComments and MNLI are out of the scope of EVaLS?

---

> > > ### Author Response · Authors · 2024-08-09
> > > **Response to Reviewer RE7s**
> > >
> > > Thank you for considering our rebuttal and paying special attention to the theoretical aspects of our work. We have tried to answer all your concerns in our response and hope you find them helpful. Please let us know if there is anything that needs to be clarified, as we find these discussions very helpful.
> > >
> > > **W3**: We would like to thank you for clarifying your arguments. We think we now understand better what you meant.
> > >
> > >
> > > We should emphasize that **both** the loss-based sampling and theoretical analysis are **class**-dependent. We state that our theoretical arguments are class-dependent in L215-218, and also L584-587 in Appendix C. Consider the setting we have assumed for the classification task and the loss function (L214 in Sec. 3.3, L583 in Appendix C). As you have stated, in the case of using logistic loss (sigmoid function), loss values are calculated as $-log(\sigma(L))$ and $-log(1-\sigma(L))$ (each for one of the classes). Therefore, the order of samples in logit and loss spaces is monotonic (L215, L584) in **each class**. The only difference between the two classes is that the direction of the monotonicity is different for them. But as our analysis are **class**-dependent, there is no worry about applying two different transformations, as the transformations themselves are **class**-dependent.
> > >
> > > Please note that our theoretical analysis aims to show that loss-based sampling can create a balanced dataset. Focusing on logit space is sufficient to demonstrate sampling thresholds. Consequently, as seen in Proposition 3.1 (and C.1), we assume that we have $1-\epsilon$ spurious correlation, and minority (proportion $\epsilon$) and majority (proportion $1-\epsilon$) samples in each class are drawn from different Gaussian distributions in the feature and hence logit spaces (as you have noted). Additional assumptions about the feature space structure (like in the paper [1] you mentioned) are more restrictive. We do not fully comprehend how assuming a more structured feature space is helpful to further develop the theory as the current assumptions are less restrictive.
> > >
> > > **W4**: Thank you. As shown in our response to reviewer 7E4z, while the mentioned methods target different subpopulation shifts or have higher supervision, our method outperforms them in almost all experiments (the only exception is Waterbirds, where some have higher WGA but lower average accuracy). We will include the new baseline results in the final paper.
> > >
> > > As stated in Sec. 2.1 (L112-L123), There exist multiple types of subpopulation shifts (see response to reviewer 7E4z for class-imbalance). Refer to reference [4] in the paper for a categorization of subpopulation shifts and their formulation (Table 1 in their paper). Our focus in designing environment-based validation (EV) is on spurious correlation. The types of subpopulation shifts in CivilComments and MultiNLI are class-imbalance and attribute-imbalance, respectively (see the definition of these datasets in Appendix D.3, L714-725).
> > >
> > > Regarding your question, please refer to L316-324. As explained there, patterns distinguishing groups in CivilComments and MultiNLI are not predictive for the target class (see tables below). This reduces their visibility in the model’s final layers (reference [29] in the paper discusses the role of various layers of neural networks in different types of shifts). Since environment inference algorithms like EIIL (see Appendix E.2 for further investigation) depend on the last layers of a trained model, they cannot infer environments with notable group shifts (defined in L195-198) in CivilComments and MultiNLI. The group shifts in CivilComments and MultiNLI (L323-324) are significantly lower than those of the datasets that are reported in Table 3. Thus, the focus of environment-based validation is on datasets with spurious correlation.
> > >
> > > Nevertheless, loss-based sampling (LS) is effective for CivilComments and MultiNLI. Our EVaLS-GL, using ground-truth group labels for model selection and loss-based sampling for retraining, outperforms all other methods on CivilComments and those with similar group supervision on MultiNLI. Only GDRO, with full group annotations during training, performs better on MultiNLI.
> > >
> > > **CivilComments and MultiNLI training sets statistics**
> > >
> > > CivilComments:
> > > |Group |Class|Attribute |# Train Data|
> > > |-|:-:|:-:|:-:|
> > > |$G_1$|0|No Identities|148186 (55%)|
> > > |$G_2$|0|Has Identities|90337 (33%)|
> > > |$G_3$|1|No Identities|12731 (5%)|
> > > |$G_4$|1|Has Identities|17784 (7%)|
> > >
> > >
> > > MultiNLI:
> > > |Group|Class|Attribute|# Train Data|
> > > |-|:-:|:-:|:-:|
> > > |$G_1$|0|No Negations|57498 (28%)|
> > > |$G_2$|0|Has Negations|11158 (5%)|
> > > |$G_3$|1|No Negations|67376 (32%)|
> > > |$G_4$|1|Has Negations|1521 (1%)|
> > > |$G_5$|2|No Negations|66630 (32%)|
> > > |$G_6$|2|Has Negations|1992 (1%)|

---

> > > > ### Comment · Reviewer_RE7s · 2024-08-10
> > > > **Reviewer Response**
> > > >
> > > > Thanks for the explanation! I think we should clarify whether your loss-base sampling approach is class-wise or not. It seems to be so from our discussion, but your pseudo-code at the top in the general rebuttal did not consider the class, and I could not verify you are selecting $|\mathcal{Y}|*2k$ samples in total from the main paper.

---

> > > > > ### Author Response · Authors · 2024-08-10
> > > > > **Clarification Regarding the Algorithm**
> > > > >
> > > > > Dear reviewer RE7s,
> > > > >
> > > > > In the general author rebuttal, we aimed to keep the pseudocode minimal and straightforward to highlight the main components of the algorithm and framework. We greatly appreciate your attention to detail in noticing that the pseudocode in the general author rebuttal did not demonstrate the class-dependency of the algorithm. Details regarding this matter have been added to the pseudo-code below.
> > > > >
> > > > > We have clarified in the paper that loss-based sampling occurs within each class. In Section 3.1, where we introduce loss-based sampling, we clearly state: *“By combining these **$2k$ samples from each class**, we construct a balanced set $D^{balanced}$”, consisting of high-loss and low-loss samples (see Figure 1(c))* (L171-172). This matches what you state in the response. Moreover, in the caption of the main figure, Figure 1(c) in the paper, it is stated: *“We evaluate train split samples on the initial ERM classifier and sort high-loss and low-loss samples of **each class** for loss-based sampling.”* Additionally, in Figures 1(c) and 1(d) in the paper, you can see that in each class, samples with the highest and lowest loss are selected.
> > > > >
> > > > > It is worth mentioning that our environment-based validation also applies to each class separately, as stated in Sec. 3.2 Partitioning Validation Set into Environments (L194-195): *“Subsequently, each environment is further divided based on sample labels, resulting in $2\times|y|$ environments.”* You can also observe that all our reported group shifts in Table 3 are class-dependent, and other environment inference methods used are class-dependent as well (L201-203).
> > > > >
> > > > > We sincerely apologize for any confusion in this matter. Please let us know if anything further needs to be clarified.
> > > > >
> > > > > -------------
> > > > >
> > > > > Class-dependency details for loss-based sampling **become** ***bold*** in the pseudo-code below.
> > > > >
> > > > > **Input:** Held-out dataset `D`, ERM-trained model `f_ERM`, maximum `k` value `maxK`
> > > > > **Output:** Optimal number of samples `k*`, Best model `f*`
> > > > > 1. Split the held-out dataset `D` into train and validation:
> > > > >    - $D_{MS}$, $D_{LL}$ = `splitDataset(D)`
> > > > > 2. Infer environments from the validation split:
> > > > >    - `envs = inferEnvs(D_MS)`
> > > > > 3. ***For each class $y\in\mathcal{Y}$***
> > > > >     - ***Sort $D_{LL}$ samples from class $y$ by their loss:***
> > > > >       - ***`sortedSamples[y] = sortByLoss(f_ERM, `$D_{LL}^y$`)`***
> > > > > 4. Initialize  `wea* = 0, k*= 0, f*= None`
> > > > > 5. For `k` from 1 to `maxK`:
> > > > >    - ***For each class $y\in\mathcal{Y}$:***
> > > > >      - Select top-`k` high-loss samples:
> > > > >        - ***`highLossSamples[y] = sortedSamples[y][:k]`***
> > > > >      - Select top-`k` low-loss samples:
> > > > >        - ***`lowLossSamples[y] = sortedSamples[y][-k:]`***
> > > > >    - Combine samples:
> > > > >      - ***`selectedSamples = {highLossSamples[y], lowLossSamples[y]} for all `$y\in\mathcal{Y}$***
> > > > >    - Retrain the last layer:
> > > > >      - `f = retrainLastLayer(selectedSamples)`
> > > > >    - Evaluate the retrained model:
> > > > >      - `wea = evaluateWEA(f, envs)`
> > > > >    - If `wea > wea*`:
> > > > >      - `wea* = wea`
> > > > >      - `f* = f`
> > > > >      - `k* = k`
> > > > > 6. return `f*`

---

> > > > > > ### Comment · Reviewer_RE7s · 2024-08-10
> > > > > > **Discussion Summary**
> > > > > >
> > > > > > Thank you for the detailed clarification on having a class-wise loss-based sampling strategy. In this case, I agree that the theoretical analysis in proposition 3.1 aligns and supports the proposed approach very well. I also apologize for not realizing the sampling strategy applies to each class, and that's why I asked for pseudocode for clarification at first. I believe the paper can be improved after including the pseudocode and more highlights to avoid confusion, which the authors have already done well in the rebuttal.
> > > > > >
> > > > > > The major limitation still lies in the experimental results. The proposed algorithm EVaLS is only somewhat better than other baselines, and the results are quite arguable when compared to SELF[2], since SELF is more flexible and outperforms EVaLS-GL on Waterbirds and CelebA. When using EIIL for validation split instead of using the ground-truth group labels, it's also limited to the case where the spurious correlations are strong and not applicable to the two NLP datasets. The authors have correctly identified and discussed these limitations.
> > > > > >
> > > > > > After checking the comments from other reviewers, there are no other obvious major concerns. I have increased my score accordingly.

---

> ### Author Response · Authors · 2024-08-11
>
> Dear Reviewer RE7s,
>
> We’re glad our clarification was helpful. It’s good to know that the pseudo-code and highlights have been effective in preventing confusion. We appreciate your suggestion to include them in the final version and will incorporate them.
>
> ### Regarding the limitations that you have stated
>
> As you mentioned, we state in the Discussion section (L356-357) that environment-based validation is limited to spurious correlation and not other types of subpopulation shifts.
>
> ### Regarding SELF [2]
>
> Please refer to the response to Reviewer 7E4z. As we described there and also in Appendix A-Related Work (L547-554), reference [2] in the paper proposes two methods:
>
> *Class-balanced (CB) last layer retraining*: See L552-554. As we discuss comprehensively in the response to Reviewer 7E4z, this method is not helpful in targeting robustness to spurious correlation, but *it improves robustness to class-imbalance, another type of subpopulation shift which is independent of spurious correlation*. For details, refer to the formulations in the response to Reviewer 7E4z.
>
> *Early-stop (ES) disagreement SELF*: As noted in the response to Reviewer 7E4z, ES disagreement requires group annotations for model selection, but not for retraining (L547-551). It also needs an early-stopped version of the trained ERM and the final model.
>
> In the following table, we restate the worst group accuracy for our methods and those of SELF on spurious correlation benchmarks. We also add the results of CB last-layer retraining and ES disagreement SELF for Dominoes-CMF.
>
> |Dataset|Group Info (Train/Val)|Waterbirds|CelebA|UrbanCars|Dominoes-CMF|
> |-|-|-|-|-|-|
> |ES disagreement (SELF)|$\text{x}/\checkmark$|$93.0_{\pm0.3}$|$83.9_{\pm0.9}$|$82.1_{\pm1.8}$|$60.5_{\pm1.3}$|
> |EVaLS-GL (ours)|$\text{x}/\checkmark$|$89.4_{\pm0.3}$|$84.6_{\pm1.6}$|$82.3_{\pm1.2}$|$63.6_{\pm1.3}$|
> |CB last-layer retraining (SELF)|$\text{x}/\text{x}$|$92.6_{\pm0.8}$|$73.7_{\pm2.8}$|$21.9_{\pm13.0}$|$53.3_{\pm5.0}$|
> |EVaLS (ours)|$\text{x}/\text{x}$|$88.4_{\pm3.1}$|$85.3_{\pm0.4}$|$82.1_{\pm0.9}$|$67.1_{\pm4.2}$|
> |ERM|$\text{x}/\text{x}$|$66.4_{\pm2.3}$|$47.4_{\pm2.3}$|$18.67_{\pm2.0}$|$50.6_{\pm1.0}$|
>
> *Waterbirds*: SELF’s methods outperform EVaLS. As SELF illustrates in their Sec. 3 (Preliminaries), since CB last-layer retraining is performed on validation data, which is group-balanced for Waterbirds, CB last-layer retraining is also group-balanced for Waterbirds. When the dataset for is not group-balanced, CB last-layer retraining achieves a worst group accuracy of $77.4_{\pm0.3}$ (Table 7 in reference [2] of the paper (SELF)).
>
> *CelebA*: As you can see, ***EVaLS and EVaLS-GL outperform both CB last-layer retraining and ES disagreement***. We believe there may have been an oversight in your response, as it incorrectly suggests their method outperforms ours. CB last-layer retraining underperforms all other benchmarked methods in Table 1.
>
> *UrbanCars*: EVaLS-GL outperforms all other methods. Both EVaLS and ES disagreement (with higher group supervision) show similar worst group accuracy. CB last-layer retraining shows no significant improvement over ERM. This is expected, as UrbanCars’ training data is class-balanced, unlike Waterbirds and CelebA. Therefore, further class-balancing for CB last-layer retraining makes no difference.
>
> *Dominoes-CMF*:
> EVaLS outperforms other methods. EVaLS-GL also outperforms methods in SELF [2]. CB last-layer retraining does not show any significant improvement compared to ERM. This is expected, as Dominoes-CMF also has class-balanced training data like UrbanCars.
>
> #### Conclusion:
>
> CB last-layer retraining targets class-imbalance shifts and is ineffective for spurious correlation. Except for the Waterbirds dataset, our methods ٍ(EVaLS and EVaLS-GL) outperform those of SELF [2] with similar group supervision on all other datasets. EVaLS shows comparable or better results (***e.g., in CelebA***) compared to methods with higher group supervision. See Table 1 in the paper for more comparisons.

---

> > ### Comment · Reviewer_RE7s · 2024-08-13
> > **Evaluation on CivilComments Dataset**
> >
> > Previously in Q3, I asked about why the performance of the proposed approach is much higher for the CivilComments dataset.
> >
> > The author responded that "The CivilComments dataset has class imbalance, so fine-tuning/retraining on class-balanced data can significantly improve group robustness."
> >
> > However, as I looked closely into the details, that might not be the case. In SELF[2], the worst-group accuracy is calculated over **4** groups by aggregating all identity categories into one spurious feature. In other benchmarks such as JTT, DFR, and AFR, the worst-group accuracy is calculated over **16** groups (8 attributes and 2 labels). Therefore, the result of EVaLS-GL in CivilComments column might not be directly comparable to each other except with SELF (added later) and the ERM/DFR implemented by SELF. Please look into this carefully and correct/rerun the misleading results if necessary.
> >
> > Relevant public GitHub repos:
> >
> > https://github.com/tmlabonte/last-layer-retraining
> >
> > https://github.com/AndPotap/afr

---

> > > ### Author Response · Authors · 2024-08-13
> > >
> > > Dear Reviewer RE7s,
> > >
> > > We deeply appreciate your notice. Our setting follows the practice in the GitHub repository `izmailovpavel/spurious_feature_learning` in `datasets.py`, which is referenced in the official implementation of DFR as an extension of their code functionality. Both (line 266 in their code and line 129 in `data/civilcomments.py` in our code) collapse grouping as “Has Identities”/”No Identities” in each class (4 groups in total). We realize that all the methods that report results around ~79-80% worst group accuracy on CivilComments (like the class-balanced schemes suggested by Reviewer 7E4z and previously reported results for EVaLS-GL) use the 4-group setting of CivilComments, while the methods (as you mentioned) that report on the 16-group setting, result in around ~69-70% worst group accuracies.
> > >
> > > We carefully considered your notice and reran the experiment of EVaLS-GL on the CivilComments dataset with 16 groups. **EVaLS-GL on CivilComments with 16 groups has worst group accuracy and average accuracy (in parentheses) of $\boldsymbol{68.0_{\pm0.5}(89.2_{\pm0.3})}$%**. Evaluating ERM on worst accuracy for 16 groups also resulted in $56.3_{\pm4.8}$ (average accuracy does not differ with the 4 group setting).
> > >
> > > We should restate that, as mentioned by previous works (reference [4]-Table 2) and as seen in the following table, CivilComments is not an instance of a dataset with spurious correlation (L313). In other words, no identity attribute in CivilComments is predictive for any of the labels (see L316-318). However, loss-based sampling with ground-truth group annotations (EVaLS-GL) still significantly improves worst group accuracy compared to initial ERM.
> > >
> > > Proportion of attributes in each class for CivilComments dataset:
> > > |Toxicity (Class)|Male|Female|LGBTQ|Christian|Muslim|Other Religions|Black|White|
> > > |-|-|-|-|-|-|-|-|-|
> > > |0|0.11|0.12|0.03|0.10|0.05|0.02|0.03|0.05|
> > > |1|0.14|0.15|0.08|0.08|0.10|0.03|0.1|0.14|
> > >
> > > ### Conclusion
> > > EVaLS-GL results in $68.0_{\pm0.5}$% worst group accuracy when CivilComments is examined with 16 groups (identity attributes in the table above in each class), and $80.5_{\pm0.4}$% when it is examined with 4 groups (“Has Identities”/”No Identities” attributes in each class).
> > >
> > > We acknowledge the discrepancy in the reported results of EVaLS-GL on the CivilComments dataset and appreciate your feedback. We hope the above explanations provide clarity for a thorough comparison. We will incorporate this discussion regarding the EVaLS-GL results on CivilComments into the final version of the paper.
> > >
> > > Finally, we again greatly appreciate your attention to this matter and your notice regarding this.
> > >
> > > ---
> > >
> > > Below, we present the updated main table (Table 1) of the paper, including the results you requested to be added to it in this rebuttal thread for a more comprehensive comparison. If a method’s worst group accuracy is better than others with a similar or weaker level of assumption about the availability of group annotations during training/validation phases, it is underlined. Worst group accuracies are bolded if a method outperforms all other methods. SC stands for spurious correlation datasets. Results for CivilComments are calculated for the 16-group setting (∗ denotes results for the 4-group setting). Other notations are similar to the main table. We would like to remind you that environment-based validation only works for spurious correlation subpopulation shift (L316-324).
> > >
> > > |Method|Group Info (Train/Val)|Waterbirds (SC)|CelebA (SC)|UrbanCars (SC)|CivilComments|MultiNLI|
> > > |-|-|-|-|-|-|-|
> > > |GDRO|$\checkmark/\checkmark$|$91.4$|$\underline{\boldsymbol{88.9}}$|$73.1_{\pm 2.0}$|$69.9$|$\underline{\boldsymbol{77.7}}$|
> > > |DFR|$\text{x}/\checkmark\checkmark$|$92.9_{\pm0.2}$|$88.3_{\pm1.1}$|$79.6_{\pm2.2}$|$\underline{\boldsymbol{70.1_{\pm0.8}}}$|$74.7_{\pm0.7}$|
> > > —-------------------------------
> > > |GDRO+EIIL|$\text{x}/\checkmark$|$77.2_{\pm1.0}$|$81.7_{\pm0.8}$|$76.5_{\pm2.6}$|$67.0_{\pm2.4}$|$61.2_{\pm0.5}$|
> > > |ES disagreement SELF|$\text{x}/\checkmark$|$\underline{\boldsymbol{93.0_{\pm0.3}}}$|$83.9_{\pm0.9}$|$82.1_{\pm1.8}$|$79.1_{\pm2.1}^*$|$70.7_{\pm2.5}$|
> > > |JTT|$\text{x}/\checkmark$|$86.7$|$81.1$|$79.5_{\pm 5.1}$|$\underline{69.3}$|$72.6$|
> > > |AFR|$\text{x}/\checkmark$|$90.4_{\pm1.1}$|$82.0_{\pm0.5}$|$80.2_{\pm2.0}$|$68.7_{\pm0.6}$|$73.4_{\pm0.6}$|
> > > |EVaLS-GL (Ours)|$\text{x}/\checkmark$|$89.4_{\pm0.3}$|$84.6_{\pm1.6}$|$\underline{\boldsymbol{82.3_{\pm1.2}}}$|$68.0_{\pm0.5}$/$80.5_{\pm0.4}^*$|$\underline{75.1_{\pm1.2}}$|
> > > —-------------------------------
> > > |EVaLS (Ours)|$\text{x}/\text{x}$|$88.4_{\pm3.1}$|$\underline{85.3_{\pm0.4}}$|$\underline{82.1_{\pm1.0}}$|Not SC|Not SC|
> > > |Class-balanced last-layer retraining|$\text{x}/\text{x}$|$\underline{92.6_{\pm0.8}}$|$73.7_{\pm2.8}$|$21.9_{\pm13.0}$|$80.4_{\pm0.8}^*$|$64.7_{\pm1.1}$|
> > > |ERM|$\text{x}/\text{x}$|$66.4_{\pm2.3}$|$47.4_{\pm2.3}$|$18.7_{\pm2.0}$|$56.3_{\pm4.8}$|$64.8_{\pm1.9}$|

---

> > > > ### Comment · Reviewer_RE7s · 2024-08-14
> > > > **Reviewer Response**
> > > >
> > > > Thank you for promptly checking and running the results for CivilComments in the correct setting. Please make sure to present a clear and fair comparison with detailed experimental setup in the revised version to avoid potential confusion and unfairness. With no other major issues left, I have increased my score accordingly.

---

> > > > > ### Author Response · Authors · 2024-08-14
> > > > >
> > > > > Dear Reviewer RE7s,
> > > > >
> > > > > Thank you for your constructive review. We’re pleased that our rebuttal successfully addressed your concerns. We ensure that the final version addresses these points to avoid any potential confusion or unfairness.
> > > > >
> > > > > Best regards,
> > > > >
> > > > > Authors of Submission 19478

---

### Author Rebuttal · Authors · 2024-08-07

We are thankful for the time and consideration that reviewers have dedicated to reviewing our work. Our work is in continuation of numerous efforts towards annotation-free group robustness (see Appendix A, L538-L549). The main contributions are as follows:

1. **Environment-based validation drops the requirement of group annotation for model selection**: RE7s (strength 3), SN9B (summary), and 7E4z (summary) point out that we use environment inference methods to achieve environments with group shifts. These environments are used for model selection via worst environment accuracy (WEA) (as described in Sec. 3.2 and Figure 1-b,d). As stated in the abstract (L15-24) and introduction (L64-69), and as our results show, for the first time, we observe that using environment inference methods to achieve environments with group shifts (as illustrated in the Appendix, Table 3) suffices for model selection to mitigate spurious correlation. Thus, we drop the requirement for the availability of group annotations for model selection, which, as we discuss in Appendix A (L549-551) is a limitation of previous efforts.

2. **Enhancing the robustness of trained models to unknown spurious correlations they rely on**: Real-world cases often involve unknown (unlabeled) spurious correlations (Abstract, L6-9), which previous methods have not adequately addressed (refer to Appendix-L538-551). Although EVaLS does not require group annotations, results show its robustness against learned shortcuts in scenarios with unknown spurious correlations. It is also effective in settings like the UrbanCars dataset, which has multiple spurious correlations that are partially known or completely unknown (SN9B, strength 2). We also propose Domino-CMF (Sec. 2.2,) as a benchmark featuring two independent spurious patterns, one of which is unknown (RE7s, summary). Notably, EVaLS, even without group supervision, achieves a higher worst group accuracy compared to methods that rely on higher levels of group supervision for the known pattern (AFR, DFR, EVaLS-GL).

3. **Providing theoretical explanations and guarantees regarding loss-based sample selection for the retraining phase**: As stated in the Introduction (L46-51), while it has been known in the literature that loss is a plausible proxy for detecting minority samples, we demonstrate (in Sec. 3.3 and more comprehensively in Appendix C) for the first time, that there are conditions for the effectiveness of loss-based sampling in mitigating spurious correlations. RE7s (strength 2) and 7E4z (strength 3) point out the analysis as a strength of our work. The insights gained from these theoretical findings help us derive loss-based sampling as an effective balancing method.

We are glad that reviewers find the paper well-written (RE7s (strength 1), SN9B (strength 1)), well-structured and presented in a clear and organized manner (7E4z (strength 1)), with a set of rich experiments and necessary theoretical explanations (SN9B (strength 1)), and also find EVaLS simple (7E4z (strength 1)), effective (7E4z (strength 1)), and efficient (RE7s (strength 4)).
## Missing Results
The missing results of Table 1 and 2 are completed (see the answer to RE7s) and will be reported in the revised version.
## Loss-Based Sampling and Model Selection Pseudocode
**Input:** Held-out dataset `D`, ERM-trained model `f_ERM`, maximum `k` value `maxK`
**Output:** Optimal number of samples `k*`, Best model `f*`
1. Split the held-out dataset `D` into train and validation:
   - `D_MS, D_LL = splitDataset(D)`
2. Infer environments from the validation split:
   - `envs = inferEnvs(D_MS)`
3. Sort `D_LL` samples by their loss:
   - `sortedSamples = sortByLoss(f_ERM, D_LL)`
4. Initialize  `wea* = 0, k*= 0, f*= None`
5. For `k` from 1 to `maxK`:
   - Select top-`k` high-loss samples:
     - `highLossSamples = sortedSamples[:k]`
   - Select top-`k` low-loss samples:
     - `lowLossSamples = sortedSamples[-k:]`
   - Combine samples:
     - `selectedSamples = {highLossSamples, lowLossSamples}`
   - Retrain the last layer:
     - `f = retrainLastLayer(selectedSamples)`
   - Evaluate the retrained model:
     - `wea = evaluateWEA(f, envs)`
   - If `wea > wea*`:
     - `wea* = wea`
     - `f* = f`
     - `k* = k`
## Sensitivity to $k$ and $l1$
The set of hyperparameters that are used could be found in Appendix D.4. Note that the parameters  $k$ (number of selected samples from each loss tail) and $\lambda$ ($l_1$ regularization factor) are selected automatically by the environment-based validation scheme depicted in the work.

You can see heatmaps for sensitivity of the worst validation group accuracy (WGA) of our method to $k$ and $\lambda$. Differences between lowest and highest WGA among all combinations for Waterbirds, CelebA, and UrbanCars are around 10%, 16%, and 25% respectively.

## Regarding Figure 2 in the paper
Note that Figure 2 illustrates the percentage of samples that are indeed minority or majority for various thresholds of x% of samples with the highest/lowest loss. Note that if n% of samples within the x% of samples with the highest/lowest loss belong to minority/majority groups of a class, the remaining (100-n)% belong to the majority/minority groups of that class.

## Distribution of logits
Below, we report characteristics of distribution of our datasets in the logit space for Waterbirds and CelebA. We use Earth Mover’s distance on logits (which is from the same unit of variable) to quantify the distance. We also report mean and std of our main datasets so that the Earth’s Mover’s Distance makes sense.
| | WaterBirds | |CelebA|
|:-:|:-:|:-:|:-:|
| |Class 1|Class 2|Class 2|
| |Min/Maj|Min/Maj|Min/Maj|
|Mean|$-6.77$/$-19.17$|$2.55/11.39$|$-1.02$/$6.42$|
|STD|$6.31$/$6.23$| $6.97$/$4.75$ |  $7.64$/$6.48$|
|Earth Mover’s Distance|$12.40$|$8.84$|$7.43$|

In addition to the table, the overall distribution of logits per group of Waterbirds and CelebA datasets is available in Figure 2 of the attached PDF.

---

### Decision · Program_Chairs · 2024-09-25

**Decision:**

Reject

**Comment:**

While the premise and approach has novelties, the overall contribution of this paper is insufficient for publication in NeurIPS. One major concern is that theory is not informative of the method: while the method proposed is a reasonably complex pipeline that combines several (existing) ideas, the theory is about balancing tails of two scalar gaussians. A theory with meaningful guarantees for the authors' proposed method would improve the paper substantially.